# Anti-Correlated Noise in Epoch-Based Stochastic Gradient Descent: Implications for Weight Variances

## Abstract

Stochastic Gradient Descent (SGD) has become a cornerstone of neural network optimization due to its computational efficiency and generalization capabilities. However, the noise introduced by SGD is often assumed to be uncorrelated over time, despite the common practice of epoch-based training where data is sampled without replacement. In this work, we challenge this assumption and investigate the effects of epoch-based noise correlations on the stationary distribution of discrete-time SGD with momentum. Our main contributions are twofold: First, we calculate the exact autocorrelation of the noise during epoch-based training under the assumption that the noise is independent of small fluctuations in the weight vector, revealing that SGD noise is inherently anti-correlated over time. Second, we explore the influence of these anti-correlations on the variance of weight fluctuations. We find that for directions with curvature of the loss greater than a hyperparameter-dependent crossover value, the conventional results for uncorrelated noise are recovered. However, for relatively flat directions, the weight variance is significantly reduced, leading to a considerable decrease in loss fluctuations compared to the constant weight variance assumption. Furthermore, we demonstrate that training with these anti-correlations enhances test performance, suggesting that the inherent noise structure induced by epoch-based training plays a crucial role in finding flatter minima that generalize better.

## 1 Introduction

Initially developed to address the challenges of computational efficiency in neural networks, stochastic gradient descent (SGD) has exhibited exceptional effectiveness in managing large datasets compared to the full gradient methods (Bottou, 1991). It has since garnered widespread acclaim in the machine learning domain (LeCun et al., 2015), with applications spanning image recognition (Krizhevsky et al., 2012; Simonyan & Zisserman, 2015; He et al., 2016), natural language processing (Sutskever et al., 2014; Brown et al., 2020), and mastering complex games beyond human capabilities (Silver et al., 2017). Alongside its numerous variants (Duchi et al., 2011; Kingma & Ba, 2015; Schmidt et al., 2021), SGD remains the cornerstone of neural network optimization.

SGD's success can be attributed to several key properties, such as rapid escape from saddle points (Ge et al., 2015) and its capacity to circumvent "bad" local minima, instead locating broad minima that generally lead to superior generalization (Hochreiter & Schmidhuber, 1997; Keskar et al., 2017; Jastrzębski et al., 2018; Smith & Le, 2018; Xie et al., 2021; Wojtowytsch, 2024). This is often ascribed to anisotropic gradient noise (Hoffer et al., 2017; Sagun et al., 2018; Zhang et al., 2018; 2019; Zhu et al., 2019; Li et al., 2020; Ziyin et al., 2022). Nonetheless, recent empirical research posits that even full gradient descent, with minor adjustments, can achieve generalization performance comparable to that of SGD (Geiping et al., 2022).

To deepen our understanding of neural network training dynamics, several studies have investigated the limiting behavior of network weights during the later stages of training (Yaida, 2019; Kunin et al., 2023). Of particular interest is the behavior of weight fluctuations near a minimum of the loss function (Mandt et al., 2017; Jastrzębski et al., 2018; Liu et al., 2021). Empirical evidence suggests that the covariance matrix $\mathbf{C}$ associated with SGD noise is proportional to the Hessian matrix $\mathbf{H}$

of the loss function (Sagun et al., 2018; Zhang et al., 2018; 2019; Zhu et al., 2019; Thomas et al., 2020; Xie et al., 2021), although this proportionality may not hold under certain conditions (Ziyin et al., 2022). Consequently, theoretical analyses predict that the stationary covariance matrix of the weights, $\Sigma$, becomes isotropic when the learning rate is sufficiently small (Jastrzębski et al., 2018; Liu et al., 2021; Kunin et al., 2023). However, recent empirical studies (Feng & Tu, 2021) have identified significant anisotropy in $\Sigma$.

In this work, we present both theoretical and empirical analyses of weight fluctuations during the later stages of training, accounting for the emergence of anti-correlations in the noise produced by SGD, which stem from the prevalent epoch-based learning schedule. As a result of these anti-correlations, we discover that the covariance matrix $\Sigma$ displays anisotropy and is smaller than expected in a subspace of weight directions corresponding to Hessian eigenvectors with small eigenvalues, while maintaining the isotropy of $\Sigma$ in directions associated with Hessian eigenvectors possessing large eigenvalues. Our theoretical predictions are validated through the analysis of a neural network's training within a subspace of its top Hessian eigenvectors.

In addition, we demonstrate that for a small convolutional network trained on CIFAR10 the anti-correlations in SGD noise described above significantly increase the test accuracy. By linking this result to a previous study on artificially added anti-correlated noise and its benefits (Orvieto et al., 2022), we argue that the anti-correlations in SGD noise suppress diffusion in flat directions, and in this way contribute to finding flatter minima with better test accuracy.

### OUR CONTRIBUTIONS

- We uncover generic anti-correlations in SGD noise that result from the common practice of drawing training examples without replacement. We calculate the autocorrelation function of SGD noise under the assumption that the noise is independent of small fluctuations in the weight vector over time. The anti-correlations arise because the noise sums to zero over an epoch where each example is presented once.

- Using the computed autocorrelation function, we develop a theory that elucidates the relationship between variances along different Hessian eigenvectors. The weight space partitions into two groups based on Hessian eigenvalues: for eigenvectors with eigenvalues exceeding a hyperparameter-dependent crossover value $\lambda_{\text{cross}}$, the isotropic variance prediction holds; for those with smaller eigenvalues, the variance is proportional to the eigenvalue and is smaller than the isotropic value. This theory accounts for the discrete nature of SGD, avoids continuous-time approximations, and incorporates heavy-ball momentum.

- For each Hessian eigenvector, there exists an intrinsic correlation time of update steps, introduced by the loss landscape and proportional to $1/\lambda_i$, where $\lambda_i$ is the corresponding Hessian eigenvalue. Additionally, there is a fixed, direction-independent noise correlation time $\tau_{\text{SGD}}$ on the order of one epoch. The smaller of these two timescales determines the actual correlation time $\tau_i$ for the update steps along a given eigenvector. Since weight variances are proportional to $\tau_i$, this leads to the emergence of two distinct variance-curvature relationships.

- We validate our theoretical predictions by analyzing the training of a neural network within a subspace spanned by its top Hessian eigenvectors. Utilizing Hessian eigenvectors offers advantages over principal component analysis of the weight trajectory, as used in a previous empirical study (Feng & Tu, 2021), where finite-size effects introduced significant artifacts.

- We investigate how the proposed weight variance affects the training error. Using a quadratic loss model, we demonstrate that noise anti-correlations significantly reduce loss fluctuations. For the distribution of the 5,000 largest Hessian eigenvalues derived from our LeNet case study, we observe a 62% reduction in loss fluctuation compared to models with constant weight variance, highlighting the importance of a broader range of Hessian eigendirections in training optimization.

- We observe that drawing examples without replacement during training leads to higher test accuracy compared to drawing with replacement. By linking our findings to a prior study (Orvieto et al., 2022) on artificially added anti-correlated noise, we propose that the higher test accuracy is connected to the anti-correlated noise inherent in the without-replacement sampling, which may encourage convergence to flatter minima.

## 2 BACKGROUND

We consider a neural network characterized by its weight vector, $\boldsymbol{\theta} \in \mathbb{R}^d$. The network is trained on a set of $N$ training examples, each denoted by $x_n$, where $n$ ranges from 1 to $N$. The loss function, defined as $L(\boldsymbol{\theta}) := \frac{1}{N} \sum_{n=1}^N l(\boldsymbol{\theta}, x_n)$, represents the average of individual losses incurred for each training example $l(\boldsymbol{\theta}, x_n)$.

It is common practice to add some kind of momentum to the SGD algorithm when training a network as it leads to faster convergence, has a smoothing effect, and helps in escaping local minima. Therefore, to keep the analysis general, we consider a training process that employs stochastic gradient descent augmented with heavy-ball momentum. This approach updates the network parameters according to the following rules:

$$\mathbf{g}_k(\boldsymbol{\theta}) = \frac{1}{S} \sum_{n \in \mathcal{B}_k} \boldsymbol{\nabla} l(\boldsymbol{\theta}, x_n) \,, \qquad \mathbf{v}_k = -\eta \mathbf{g}_k(\boldsymbol{\theta}_{k-1}) + \beta \mathbf{v}_{k-1} \,, \qquad \boldsymbol{\theta}_k = \boldsymbol{\theta}_{k-1} + \mathbf{v}_k \,. \quad (1)$$

Here, $k$ signifies the discrete update step index, $\eta$ is the learning rate, and $\beta$ is the momentum parameter. The stochastic gradient at each step is computed with respect to a batch of $S \ll N$ random examples. Each batch is denoted by $\mathcal{B}_k = \{n_1, \ldots, n_S\}$, where $n_j \in \{1, \ldots, N\}$. The training process is structured into epochs. During each epoch, every training example is used exactly once, implying that the examples are drawn without replacement and do not recur within the same epoch.

In the realm of SGD as opposed to full gradient descent, we introduce noise, denoted as $\boldsymbol{\delta} \mathbf{g}_k(\boldsymbol{\theta}) := \mathbf{g}_k(\boldsymbol{\theta}) - \boldsymbol{\nabla} L(\boldsymbol{\theta})$, with a covariance matrix $\mathbf{C}(\boldsymbol{\theta}) := \mathrm{cov}\big(\boldsymbol{\delta} \mathbf{g}_k(\boldsymbol{\theta}), \boldsymbol{\delta} \mathbf{g}_k(\boldsymbol{\theta})\big)$. The noise covariance matrix is known to be proportional to the gradient sample covariance matrix $\mathbf{C_0}(\boldsymbol{\theta}) := \frac{1}{N-1} \sum_{n=1}^N \boldsymbol{\nabla} \left[ l(\boldsymbol{\theta}, x_n) - L(\boldsymbol{\theta}) \right] \boldsymbol{\nabla}^\top \left[ l(\boldsymbol{\theta}, x_n) - L(\boldsymbol{\theta}) \right]$. A detailed derivation of this relation, particularly for the case of drawing without replacement, is provided in Appendix D.

As in previous studies, our primary theoretical focus is on the asymptotic – or limiting – covariance matrix of the weights, denoted by $\boldsymbol{\Sigma} := \mathrm{cov}\big(\boldsymbol{\theta}_k, \boldsymbol{\theta}_k\big)$. This means we are interested in the covariance computed over an infinite run of SGD optimization, rather than the covariance at a fixed update step $k$ computed over multiple runs of SGD (see Appendix A).

To better understand the behavior of the weight variances, we further examine the covariance matrix of the velocities, $\boldsymbol{\Sigma}_\mathbf{v} := \mathrm{cov}\big(\mathbf{v}_k, \mathbf{v}_k\big)$. We then explore the ratio of the weight variance to the velocity variance in any given direction, which we denote as $\tau_i$. Under general assumptions, this ratio equates to the velocity correlation time of the corresponding direction (see Section 4.2).

## 3 RELATED WORK

### 3.1 HESSIAN AND GRADIENT SAMPLE COVARIANCE

The assumption of a strong alignment between the gradient sample covariance $\mathbf{C_0}$ and the Hessian matrix of the loss function $\mathbf{H}$ is widely used in the literature (Jastrzębski et al., 2018; Zhang et al., 2019; Liu et al., 2021). Various theoretical arguments suggest that when a neural network's output closely matches the example labels, these two matrices should be similar (Jastrzębski et al., 2018; Sagun et al., 2018; Zhang et al., 2018; Zhu et al., 2019; Martens, 2020). However, as highlighted by Thomas et al. (2020), even slight deviations between network predictions and labels can theoretically disrupt this relationship. Nevertheless, empirical observations often reveal a strong alignment between the gradient sample covariance and the Hessian matrix near a minimum.

Zhang et al. (2019) investigated this assumption numerically. They analyzed both matrices in a specific basis that exhibits varying curvature across different directions. Their findings showed a close correspondence between curvature and gradient variance along a given direction in a convolutional image recognition network, and a reasonably good relationship in a transformer model. Similarly, Thomas et al. (2020) provided both theoretical arguments and empirical evidence across different image recognition architectures. While they did not find an exact match between the two matrices, they observed a proportionality indicated by a high cosine similarity. Xie et al. (2021) also explored this relationship using an image recognition network. In the eigenspace of the Hessian

matrix, they plotted entries within a specific interval against corresponding entries from the gradient sample covariance and observed a close match.

Ziyin et al. (2022) emphasized the importance of considering the conditions under which the approximation of proportionality between the two matrices is valid, noting that the approximation may be questionable otherwise. However, the conditions they specify – particularly being near a minimum with low loss – are consistent with our empirical setup (see Section 5.1 and Appendix H).

### 3.2 Limiting Dynamics and Weight Fluctuations

Several studies have examined the limiting dynamics of SGD, often modeling it as a stochastic differential equation (SDE). Researchers such as Mandt et al. (2017) and Jastrzębski et al. (2018) commonly approximate the loss near a minimum as a quadratic function, representing the SDE as a multivariate Ornstein-Uhlenbeck (OU) process. This process suggests a stationary weight distribution with Gaussian fluctuations. Jastrzębski et al. (2018) further assume that the gradient covariance is proportional to the Hessian, observing under these conditions that the weight fluctuations are isotropic. Kunin et al. (2023), who also incorporate momentum into their analysis, predict and empirically verify isotropic weight fluctuations. Chaudhari & Soatto (2018) investigate the SDE without assuming a quadratic loss or equilibrium, gaining insights via the Fokker-Planck equation.

Alternatively, some studies derive relationships from a stationarity assumption rather than a continuous-time approximation (Yaida, 2019; Liu et al., 2021; Ziyin et al., 2022). Yaida (2019) assumes that the weight trajectory follows a stationary distribution and derives general fluctuation-dissipation relations from this premise. Liu et al. (2021) go further by assuming a quadratic loss function, enabling them to derive exact relations for the weight variance of SGD with momentum. If the gradient covariance is additionally assumed to be proportional to the Hessian, their results also predict that the weight variance is approximately isotropic, except in directions where the product of the learning rate and the Hessian eigenvalue is significantly large.

These computed weight variances are explicitly applied in various contexts, such as calculating the escape rate from a minimum or assessing the approximation error in SGD – which captures the additional training error attributed to noise (Liu et al., 2021).

Feng & Tu (2021) present a phenomenological theory based on their empirical findings, which, unlike Kunin et al. (2023), also accounts for flat directions. They describe a general inverse variance-flatness relationship by analyzing the weight trajectories of different image recognition networks via principal component analysis. They discovered a power law relationship between the curvature of the loss and the weight variance $\sigma_{\theta,i}^2$ in any given direction, where higher curvature corresponds to higher variance. They also observed that both the velocity variance $\sigma_{v,i}^2$ and the correlation time $\tau_i$ are larger for higher curvatures.

In our approach, we avoid the continuous-time approximation and instead base our results on the assumption that the weights adhere to a stationary distribution near a quadratic minimum.

## 4 Theory

### 4.1 Autocorrelation of the noise

We consider the correlation between two noise terms arising from different SGD update steps. Specifically, when examples are sampled without replacement while keeping the weight vector $\boldsymbol{\theta}$ constant, inherent anti-correlations emerge in the noise. Below, we provide a conceptual motivation for this phenomenon and refer the reader to Appendix D for a detailed derivation.

Suppose the ratio $M := N/S$ is an integer, where $N$ is the total number of examples and $S$ is the batch size. This ensures that each epoch consists of $M$ batches, with every example presented exactly once per epoch. For a fixed weight vector $\boldsymbol{\theta}$, the mean of the gradients computed over one epoch, $\mathbf{g}_k(\boldsymbol{\theta})$, equals the total gradient $\boldsymbol{\nabla} L(\boldsymbol{\theta})$. Consequently, the sum of the noise components $\boldsymbol{\delta}\mathbf{g}_k(\boldsymbol{\theta}) = \mathbf{g}_k(\boldsymbol{\theta}) - \boldsymbol{\nabla} L(\boldsymbol{\theta})$ over one epoch must be zero. This implies that if a noise term points in a particular direction at the beginning of an epoch, subsequent noise terms within the same epoch are constrained to partially cancel it out, leading to anti-correlations.

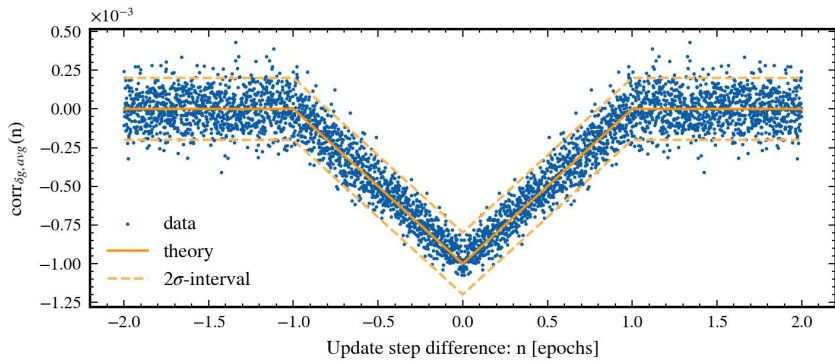

Figure 1: Autocorrelations of the SGD noise observed over a span of 20 epochs, equivalent to 20,000 update steps. This data is collected from a later phase in the training process. The autocorrelation is projected onto 5,000 Hessian eigenvectors, and the result is averaged. The theoretical prediction Equation (2) is also displayed along with a $2\sigma$-interval, where $\sigma$ represents the expected standard deviation of the SGD noise. The zero-point correlation is omitted as it is inherently equal to one.

Therefore, the anti-correlation between two noise terms within the same epoch can be expressed as a negative correlation proportional to the equal-time correlation, scaled by a factor of $\frac{1}{M-1}$, where $M$ is the number of batches per epoch. Next, we consider the probability that two batches $k$ and $k + h$, separated by $h$ update steps, belong to the same epoch. This probability is given by $\frac{M-|h|}{M}$ for $|h| \leq M$, and zero otherwise. Combining this with the anti-correlation derived above, we arrive at the correlation formula:

**Theorem 4.1.** *If the total number of examples $N$ is an integer multiple of the batch size $S$ and the parameters $\boldsymbol{\theta}$ of a network are kept fixed, then the autocorrelation formula for the gradient noise of an epoch-based learning schedule, where the examples for one epoch are drawn without replacement, is given by*

$$\mathrm{cov}\big(\boldsymbol{\delta}\mathbf{g}_k(\boldsymbol{\theta}), \boldsymbol{\delta}\mathbf{g}_{k+h}(\boldsymbol{\theta})\big) = \mathbf{C}(\boldsymbol{\theta}) \cdot \left( \delta_{h,0} - \mathbf{1}_{\{1,\dots,M\}}(|h|)\frac{M - |h|}{M(M - 1)} \right) , \tag{2}$$

*where $M := N/S$ signifies the number of batches per epoch.*

In the above theorem $\mathbf{1}_A(k)$ represents the indicator function over the set A, which is one for $k \in A$ and zero otherwise and $\delta_{i,j}$ represents the Kronecker delta. The actual noise autocorrelation is illustrated in Figure 1, with the experimental details elaborated in Section 5.2. The complete calculation is available in Appendix D, highlighting the relationship $\mathbf{C}(\boldsymbol{\theta}) = (1/S) \cdot (1 - S/N) \cdot \mathbf{C_0}(\boldsymbol{\theta})$ with the gradient sample covariance matrix $\mathbf{C_0}$.

Strictly speaking, the formula derived above applies only to a static weight vector. During training, the weights change with each update step, potentially altering this relationship. However, if the weights remain relatively constant over one epoch, this correlation approximately holds. Empirically, we find that in the later stages of training, the theoretical prediction given by Equation (2) closely matches the actual training dynamics, as evidenced by the strong agreement with the data shown in Figure 1.

When we sample the examples with replacement during training, there are no anti-correlations (see Appendix G). Further numerical investigations show that also in the case where $N$ is not an integer multiple of $S$, the later derived relations for the variances still hold (see Appendix I).

## 4.2 CORRELATION TIME DEFINITION

To gain a more comprehensive understanding of the weight variance behavior, we additionally examine the velocity variance and the ratio between them. We label this ratio, scaled by a factor of two, as the correlation time $\tau_i$ (see Equation (3)). This definition aligns with that of the velocity

correlation time, hence justifying the nomenclature. The equivalence stands under general assumptions, such as a sufficiently fast decay of the weight correlations, $\text{cov}(\boldsymbol{\theta}_k, \boldsymbol{\theta}_{k+n})$, which are satisfied in our problem setup described in Section 4.3 (see Appendix B.4), where the weight correlations will even decay exponentially fast.

**Theorem 4.2.** *Under general assumptions, stated in Appendix C and satisfied by the problem setup described in Section 4.3, it holds that*

$$\tau_i := \frac{2\sigma_{\theta,i}^2}{\sigma_{v,i}^2} = \frac{\sum_{n=1}^{\infty} n \cdot \text{cov}(v_{k,i}, v_{k+n,i})}{\sum_{n=1}^{\infty} \text{cov}(v_{k,i}, v_{k+n,i})} \,, \tag{3}$$

*justifying the label correlation time for this variance ratio.*

Here, $\sigma_{\theta,i}^2$ and $\sigma_{v,i}^2$ represent the variances of the weights and velocities, respectively, in a given direction $\mathbf{p}_i$, where $\theta_{k,i} := \boldsymbol{\theta}_k \cdot \mathbf{p}_i$ and $v_{k,i} := \mathbf{v}_k \cdot \mathbf{p}_i$. If the covariance matrices $\boldsymbol{\Sigma}$ and $\boldsymbol{\Sigma}_{\mathbf{v}}$ commute, they can be simultaneously diagonalized, allowing us to choose shared eigenvectors $\mathbf{p}_i$. In this case, $\sigma_{\theta,i}^2$ and $\sigma_{v,i}^2$ become the corresponding eigenvalues. The detailed derivation is presented in Appendix C.

### 4.3 VARIANCE FOR LATE TRAINING PHASE

Given the autocorrelation of the noise calculated earlier, we aim to present the expected variances of the weights and velocities during the later stages of training. To characterize the conditions of this phase, we make the following assumptions.

**Assumption 1: Quadratic approximation** We assume that we have reached a minimum point of the loss function, which can be adequately represented with a quadratic form as $L(\boldsymbol{\theta}) = L_0 + \frac{1}{2}(\boldsymbol{\theta} - \boldsymbol{\theta}_*)^\top \mathbf{H}(\boldsymbol{\theta} - \boldsymbol{\theta}_*)$. Without loss of generality, we set $L_0 = 0$ and $\boldsymbol{\theta}_* = 0$, simplifying the expression to $L(\boldsymbol{\theta}) = \frac{1}{2}\boldsymbol{\theta}^\top \mathbf{H}\boldsymbol{\theta}$ .

**Assumption 2: Anti-correlated noise** We assume that the covariance of the SGD noise is static and that its autocorrelation follows the relation previously calculated in Equation (2), even when the weight vector is not static.

While the noise generally exhibits state dependence, it is reasonable to assume a constant covariance during the later stages of training and within the time frame of our analysis. As described by Ziyin et al. (2022), state dependence enters the noise through the value of the training loss. Since we observe only minimal changes in the loss during our analysis period (see Appendix H), assuming a static noise covariance is justified.

**Assumption 3: Hessian and noise covariance commute** We assume that the covariance of the noise commutes with the Hessian matrix, $[\mathbf{C}, \mathbf{H}] = 0$.

This assumption is not strictly necessary, but it simplifies the analysis. The theory predicts a variance reduction in the eigenspace of the Hessian with relatively small eigenvalues without requiring commutativity (see Appendix L). However, when $[\mathbf{C}, \mathbf{H}] \neq 0$, the calculated weight and velocity variances are no longer exact eigenvalues of their respective covariance matrices; instead, they represent variances along the directions of the Hessian eigenvectors. As long as $\mathbf{C}$ and $\mathbf{H}$ approximately commute – as discussed in Section 3.1 – these variances provide a good approximation of the actual eigenvalues. For further discussion, we refer to Appendix L, where we demonstrate that although $[\mathbf{C}, \mathbf{H}] = 0$ is not strictly satisfied in our empirical investigation, the eigenbasis of $\mathbf{H}$ remains a reasonable approximation for the eigenbasis of $\mathbf{C}$. Moreover, due to finite sample sizes, the actual eigenvalues of the empirically recorded weight covariance matrix $\boldsymbol{\Sigma}$ are inherently skewed. Therefore, analyzing the weight variance in the eigenbasis of $\mathbf{H}$ is beneficial, as we explain in Appendix E.

Additionally, we assume that $0 \leq \beta < 1$ and $0 < \eta\lambda_i < 2(1+\beta)$ for all eigenvalues $\lambda_i$ of $\mathbf{H}$. If these conditions are not met, the weight fluctuations would diverge.

**Calculation for one eigenvalue** With the previously stated assumptions in place, the covariance matrices $\boldsymbol{\Sigma}$ and $\boldsymbol{\Sigma}_{\mathbf{v}}$ commute with $\mathbf{C}$, $\mathbf{H}$, and with each other (see Appendix B.1). As a result, they all share a common eigenbasis $\mathbf{p_i}$, with $i = 1, \ldots, d$, which facilitates the computation of

the expected variance. We will outline the most important steps here, while details can be found in Appendix B.2. For a given common eigenvector $\mathbf{p_i}$ we project the relevant variables onto this vector. This yields the projected weight $\theta_{k,i} := \mathbf{p_i} \cdot \boldsymbol{\theta}_k$, velocity $v_{k,i} := \mathbf{p_i} \cdot \mathbf{v}_k$, and noise term $\delta g_{k,i} := \mathbf{p_i} \cdot \boldsymbol{\delta g}_k$ at the update step $k$. Correspondingly, we define the eigenvalues for the common eigenvector $\mathbf{p_i}$ as $\lambda_i$ for $\mathbf{H}$, $\sigma_{\theta,i}^2$ for $\boldsymbol{\Sigma}$, $\sigma_{v,i}^2$ for $\boldsymbol{\Sigma_v}$, and $\sigma_{\delta g,i}^2$ for $\mathbf{C}$. We denote the number of batches per epoch as $M = N/S$, presuming it is an integer.

By introducing the vector $\mathbf{x}_{k,i} := \begin{pmatrix} \theta_{k,i} & \theta_{k-1,i} \end{pmatrix}^\top$ that contains not only the current weight variable but also the weight variable with a one-step time lag we can write the update equation as:

$$\mathbf{x}_{k,i} = \mathbf{D_i}\mathbf{x}_{k-1,i} - \eta \delta g_{k,i}\mathbf{e}_1 \ , \tag{4}$$

where $\mathbf{e}_1 := \begin{pmatrix} 1 & 0 \end{pmatrix}^\top$ and the matrix $\mathbf{D_i}$ governs the deterministic part of the update. Together with its explicit expression, we further define a correlation term:

$$\mathbf{D_i} := \begin{pmatrix} 1 + \beta - \eta\lambda_i & -\beta \\ 1 & 0 \end{pmatrix} \ , \quad \mathbf{E_i} := \mathbf{D_i}\,\mathrm{cov}\left(\mathbf{x}_{k-1,i}, \delta g_{k,i}\mathbf{e}_1\right) \ . \tag{5}$$

The term $\mathbf{E_i}$ encapsulates the correlation between the current weight variable and the noise term of the next update step. Typically, noise terms are assumed to be temporally uncorrelated, which would render $\mathbf{E_i}$ null. However, given the anti-correlated nature of the noise, we find a non-zero $\mathbf{E_i}$. An explicit expression of $\mathbf{E_i}$ can be found in Appendix B.2.

**Theorem 4.3.** *With the above assumptions and definitions, the following relation for the weight and velocity variances holds:*

$$\begin{pmatrix} \sigma_{\theta,i}^2 \\ \sigma_{v,i}^2 \end{pmatrix} = \eta^2 \sigma_{\delta g,i}^2 \mathbf{F_i} \left[ \mathbf{e}_1 - \left( \mathbf{E_i} + \mathbf{E_i}^\top \right) \mathbf{e}_1 \right] \ , \tag{6}$$

*where the matrix $\mathbf{F_i}$ is explicitly expressed as:*

$$\mathbf{F_i} = \frac{1}{\left(1 - \beta\right)\left(2(1 + \beta) - \eta\lambda_i\right)} \begin{pmatrix} \frac{1+\beta}{\eta\lambda_i} & \frac{2\beta(\eta\lambda_i - 1 - \beta)}{\eta\lambda_i} \\ 2 & 2(\eta\lambda_i - 2) \end{pmatrix} \ . \tag{7}$$

The calculations can be found in Appendix B.2. The exact relation Equation (6) can be easily evaluated numerically but it can also be approximated by assuming that $M \gg 1/(1 - \beta)$, which implies that the correlation time induced by momentum is substantially shorter than one epoch. Consequently, two distinct regimes of Hessian eigenvalues emerge, separated by $\lambda_{\mathrm{cross}} := 3(1 - \beta)/(\eta M)$. For each of these regimes, specific simplifications apply. Notably, at $\lambda_{\mathrm{cross}}$, both approximations converge.

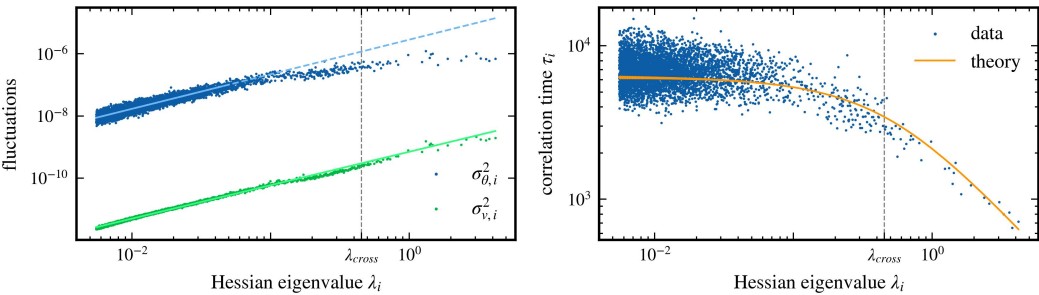

Figure 2: Relationship between Hessian eigenvalues and the variances of weights and velocities, as well as correlation times. The mean velocity of the weight trajectory has been subtracted (see Section 5.3). In the left panel, we present the variances of weights and velocities. The solid lines signify the regions utilized for a linear fit. The exponents resulting from the power law relationship are $1.077 \pm 0.012$ for weight variance and $1.066 \pm 0.002$ for velocity variance, with a $2\sigma$-error. Our theory suggests these exponents should be equal to one. The right panel showcases the correlation time together with the theoretical prediction resulting from Equation (6).

**Corollary 4.4. Relations for large Hessian eigenvalues:** *For Hessian eigenvectors with eigenvalues $\lambda_i > \lambda_{cross}$ and when $M \gg 1/(1 - \beta)$, the effects of noise anti-correlations are minimal. Consequently, we can use the following approximate relationships, which also hold true in the absence of correlations and, therefore, also when drawing examples with replacement:*

$$\sigma_{\theta,i}^2 \approx \frac{\eta^2 \sigma_{\delta g,i}^2}{(1-\beta)\big(2(1+\beta) - \eta\lambda_i\big)} \cdot \frac{1+\beta}{\eta\lambda_i} \; , \; \sigma_{v,i}^2 \approx \frac{\eta^2 \sigma_{\delta g,i}^2}{(1-\beta)\big(2(1+\beta) - \eta\lambda_i\big)} \cdot 2 \; , \tag{8}$$

*and $\tau_i \approx \frac{1+\beta}{\eta\lambda_i}$ . The detailed derivation of these formulas is presented in Appendix B.3.*

It is frequently the case that the product $\eta\lambda_i$ is considerably less than one, enabling us to further simplify the prefactor of the variances. Assuming the noise covariance matrix is proportional to the Hessian matrix, such that $\sigma_{\delta g,i}^2 \propto \lambda_i$, we derive the following power laws for the variances: $\sigma_{\theta,i}^2 \propto \text{const.}$ and $\sigma_{v,i}^2 \propto \lambda_i$. As a result, in the subspace spanned by Hessian eigenvectors with eigenvalues $\lambda_i > \lambda_{\text{cross}}$, our theory predicts an isotropic weight variance $\mathbf{\Sigma}$.

**Corollary 4.5. Relations for small Hessian eigenvalues:** *In the case of Hessian eigenvectors associated with eigenvalues $\lambda_i < \lambda_{cross}$ and under the condition that $M \gg 1/(1 - \beta)$, the noise anti-correlation significantly modifies the outcome. We can express the approximate relationships as follows:*

$$\sigma_{\theta,i}^2 \approx \frac{\eta^2 \sigma_{\delta g,i}^2}{2(1-\beta)(1+\beta)} \cdot \frac{M}{3}\frac{1+\beta}{1-\beta} \; , \; \sigma_{v,i}^2 \approx \frac{\eta^2 \sigma_{\delta g,i}^2}{2(1-\beta)(1+\beta)} \cdot 2 \tag{9}$$

*and $\tau_i \approx \frac{M}{3}\frac{1+\beta}{1-\beta} =: \tau_{SGD}$ . Therefore, the weight variance is reduced by a factor of $\frac{M\eta\lambda_i}{3(1-\beta)}$ compared to the case without anti-correlations. The derivation of these formulas is provided in Appendix B.3.*

If we once again assume that the noise covariance matrix is proportional to the Hessian matrix, such that $\sigma_{\delta g,i}^2 \propto \lambda_i$, we obtain the following power laws for the variances: $\sigma_{\theta,i}^2 \propto \lambda_i$ and $\sigma_{v,i}^2 \propto \lambda_i$. This implies that in the subspace spanned by Hessian eigenvectors with eigenvalues $\lambda_i < \lambda_{\text{cross}}$, the weight variance $\mathbf{\Sigma}$ is not isotropic but proportional to the Hessian matrix $\mathbf{H}$. It is noteworthy that the correlation time $\tau_i \approx \frac{M}{3}\frac{1+\beta}{1-\beta} =: \tau_{\text{SGD}}$ in this subspace is independent of the Hessian eigenvalue. Moreover, $\tau_{\text{SGD}}$ is equivalent to the correlation time of the noise $M/3$, up to a factor that depends on momentum.

## 5 NUMERICS

### 5.1 ANALYSIS SETUP

In order to corroborate our theoretical findings, we have conducted a small-scale experiment. We have trained a LeNet architecture, similar to the one described in (Feng & Tu, 2021), using the CIFAR10 dataset (Krizhevsky, 2009). LeNet is a compact convolutional network comprised of two convolutional layers followed by three dense layers. The network comprises approximately 137,000 parameters. Here we present results for a single seed and specific hyperparameters. However, we have also performed tests with different seeds and combinations of hyperparameters, all of which showed comparable qualitative behavior (see Appendix I). Furthermore, in Appendix K we studied a ResNet architecture (He et al., 2016), a more modern network, where we obtained similar results.

The training parameters together with the schedule are described in Appendix H and result in a thousand minibatches per epoch, $M = 1,000$. The setup achieves 100% training accuracy and 63% testing accuracy. We compute the variances right after the initial schedule over a period of 20 additional epochs, equivalent to 20,000 update steps. Throughout this analysis period, a constant learning rate is maintained and the recorded weights are designated by $\boldsymbol{\theta}_k$, with $k = K, \ldots, K + T$ and $T = 20,000$. Given the impracticability of obtaining the full covariance matrix for all weights over this period due to the excessive memory requirements, we limit our analysis to a specific subspace. Employing the resource-efficient Pearlmutter trick (Pearlmutter, 1994), we approximate the 5,000 largest eigenvalues and their associated eigenvectors of the Hessian matrix $\mathbf{H}(\boldsymbol{\theta}_K)$ at the beginning of the analysis period, drawn from the roughly 137,000 total. The eigenvectors of the Hessian matrix are represented by $\mathbf{p_i}$, and the projected weights by $\theta_{k,i} = \boldsymbol{\theta}_k \cdot \mathbf{p_i}$. The variances are computed exclusively for these particular directions. The distribution of the approximated 5,000 eigenvalues is illustrated in Appendix F.

## 5.2 Noise Autocorrelations

We scrutinize the correlations of noise by recording both the minibatch gradient $\mathbf{g}_k(\boldsymbol{\theta}_k)$ and the total gradient $\nabla L(\boldsymbol{\theta}_k)$ at each update step throughout the analysis period, enabling us to capture the actual noise term $\boldsymbol{\delta}\mathbf{g}_k(\boldsymbol{\theta}_k)$. All these are projected onto the approximated Hessian eigenvectors. The theoretical prediction for the anti-correlation of the noise is proportional to the inverse of the number of batches per epoch, which, in our case, is on the order of $10^{-3}$. To extract the predicted relationship from the fluctuating data, we compute the autocorrelation of the noise term for each individual Hessian eigenvector. We then proceed to average these results across the 5,000 approximated eigenvectors. Figure 1 provides a visual representation of this analysis, showcasing a strong alignment between the empirical autocorrelation of noise and the prediction derived from our theory.

## 5.3 Variances and Correlation time

Previous studies have observed that network weights continue to traverse the parameter space even after the loss appears to have stabilized (Hoffer et al., 2017; Feng & Tu, 2021; Kunin et al., 2023). This behavior persists despite the use of L2 regularization and implies that the recorded weights, $\boldsymbol{\theta}_k$, do not settle into a stationary distribution. Notably, however, over the course of the 20 epochs under scrutiny, the weight movement, excluding the SGD noise, appears to be approximately linear in time. This suggests that the mean velocity $\bar{\mathbf{v}} := \langle \mathbf{v}_k \rangle$ is substantial compared to the SGD noise. To isolate this ongoing movement and uncover the underlying structure, we redefine $\boldsymbol{\theta}_k$ and $\mathbf{v}_k$ by subtracting the mean velocity. This results in $\boldsymbol{\theta}_k^{(\mathrm{s})} := \boldsymbol{\theta}_k - \bar{\mathbf{v}} \cdot k$ and $\mathbf{v}_k^{(\mathrm{s})} := \mathbf{v}_k - \bar{\mathbf{v}}$. We then compute the variances of these redefined values, $\boldsymbol{\theta}_k^{(\mathrm{s})}$ and $\mathbf{v}_k^{(\mathrm{s})}$, which exhibit a more stationary distribution.

Again, we limit our variance calculations to the directions of the 5,000 approximated Hessian eigenvectors. In the two different regimes of Hessian eigenvalues, either greater or lesser than the crossover value $\lambda_{\mathrm{cross}}$, the weight and velocity variance closely follow the respective power law predictions from our theory (see left panel of Figure 2). The slight discrepancy, where the predicted exponent of one does not lie within the error bars, may arise from minor deviations in the noise covariance from the approximation $\mathbf{C} \propto \mathbf{H}$. The calculated correlation time, derived from the ratio between the weight and velocity variance, aligns reasonably well with our theoretical predictions (see right panel of Figure 2). This correlation time prediction remains independent of the exact relation between $\mathbf{C}$ and $\mathbf{H}$, thereby providing a more general result.

# 6 Limitations

As highlighted in Section 5.3, the weight distribution is not entirely stationary; the weights exhibit a finite mean drift. To fully understand the underlying weight variances, it is essential to account for this mean velocity. However, for time windows extending beyond the measured 20 epochs, the mean velocity's variability increases. This change can cause the average weight movement to deviate from linearity, potentially leading to higher variances in flat directions than predicted.

In general, the late phase of training under scrutiny may not be representative of the entire training process. However, we argue that exploring such a regime, especially where a quadratic approximation of the loss seems feasible, can provide important insights into the dynamics of SGD, and in particular its stability around a minimum. Furthermore, it is an important starting point for our research, since such a situation can be treated analytically.

# 7 Discussion

We provide an intuitive interpretation of our results by examining the correlation times associated with different Hessian eigenvectors, which we also corroborate empirically. Although the 20-epoch analysis period may influence the observed correlation times – since it is comparable to the maximum predicted correlation time $\tau_{\mathrm{SGD}}$ – we observe considerable differences when we sample examples without replacement during training (see Appendix G). This suggests that the observed correlation time behavior arises from the correlations introduced by sampling without replacement.

Throughout our analysis, we primarily focus on the weight covariance matrix $\boldsymbol{\Sigma}$, considering the correlation time $\tau_i$ and the velocity covariance $\boldsymbol{\Sigma}_{\mathbf{v}}$ as auxiliary variables. Our results indicate that

the velocity covariance $\Sigma_{\mathbf{v}}$ is directly proportional to the noise covariance $\mathbf{C}$. However, the behavior of the weight covariance $\Sigma$ is more intricate due to the autocorrelation time, which connects the two variances. Specifically, for a single eigendirection, this relationship is given by $\sigma_{\theta,i}^2 = \frac{1}{2}\tau_i \sigma_{v,i}^2$. Examining these quantities in detail provides further insights. Moreover, our findings for these three quantities differ from those reported in a previous empirical study (Feng & Tu, 2021). In Appendix E, we explain how the different analysis methods used in that study led to results impacted by finite-size effects.

In this manuscript, we examine the influence of smaller Hessian eigenvalues, which are often overlooked in discussions of minima characteristics in optimization landscapes. While prior work has focused on larger eigenvalues, we highlight the impact of reduced weight variance in flat directions. This observed reduction may help explain why SGD exploration predominantly occurs within a limited subspace of Hessian eigenvectors with higher curvature, as reported in previous studies (Gur-Ari et al., 2018; Xie et al., 2021).

To further substantiate this argument, we examine the impact of weight variance on the additional training error. In a simplified quadratic loss model, the expected loss variance for a Hessian eigenvalue $\lambda_i$ and Gaussian weight fluctuations $\sigma_{\theta,i}^2$ is $l_{\text{fluct},i} = \frac{1}{2}\lambda_i \sigma_{\theta,i}^2$. Although an individual flat direction contributes relatively little to the total loss fluctuation $l_{\text{fluct}} = \sum_i l_{\text{fluct},i}$, this is counterbalanced by their abundance. By analyzing the 5,000 largest Hessian eigenvalues of LeNet, our model predicts a 62% reduction in loss fluctuation compared to assuming constant weight variance. This effect would be even stronger if all 137,000 directions were considered.

Furthermore, we argue that flat directions are crucial for generalization (see Appendix J). By decomposing LeNet's network weights into the Hessian eigenbasis, we find that omitting the projection onto the top 35 eigenvectors only reduces test accuracy from 63% to 54%, while excluding the top 5,000 eigenvectors drops it to 44%. Accuracy falls to 10% – equivalent to random guessing – only after discarding approximately 7,000 eigenvectors. This demonstrates the importance of small Hessian eigenvalues and weight fluctuations in flat directions for both loss reduction and generalization.

To further assess the impact of anti-correlations in the gradient noise, we investigated the difference in generalization performance between drawing batches in SGD with and without replacement. We trained the network described in Section 5.1 using the same training schedule and 20 different random seeds, considering the maximum test accuracy computed after each epoch. We found that training without replacement yielded test accuracies that were 0.8% ± 0.2% higher than training with replacement. Specifically, the maximum test accuracy for SGD without replacement was 64.5% on average, compared to 63.7% for SGD with replacement.

We relate this result directly to the anti-correlations we have described by comparing it with a prior study by Orvieto et al. (2022). In that study, the authors considered full-batch gradient descent with artificially added noise that is anti-correlated in time. This noise was found to be beneficial for test accuracy and led to flatter minima. While the anti-correlations they considered have a very short correlation time, they are otherwise very similar to those inherent in SGD without replacement. We therefore propose that the positive effects described in their study could be extended to SGD without replacement due to these anti-correlations.

**Conclusion**

Our investigation of anti-correlations in SGD noise – arising from drawing examples without replacement – reveals a lower-than-expected weight variance in Hessian eigendirections with eigenvalues smaller than the crossover value $\lambda_{\text{cross}}$. By introducing the concepts of intrinsic correlation time, shaped by the loss landscape, and a constant noise correlation time, we provide deeper insights into the dynamics of SGD optimization. The reduced weight variance may allow gradients in flat directions to dominate fluctuations, steering the network toward even flatter minima. Generally, training with anti-correlated noise leads to improved generalization performance, suggesting that our findings explain an important property of SGD that contributes to its success.

## REPRODUCIBILITY STATEMENT

We have made significant efforts to ensure the reproducibility of our work. The code for performing all experiments described in the manuscript, as well as for generating all figures, is provided in the

supplementary material. All assumptions related to the theoretical analysis are clearly stated in the main text, and detailed working out of the theoretical considerations can be found in the Appendix.

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

## A  DEFINITION OF LIMITING QUANTITIES

When we speak of a covariance matrix or an average in the main text and in the following sections of the appendix, we mean the limiting average or the limiting covariance, unless otherwise specified. In other words, we are interested in the average of a quantity over one infinite run of SGD optimization, not the mean value for a fixed update step $k$ averaged over multiple runs of SGD optimization. With this in mind, we define the covariance matrix of two quantities $\mathbf{a}_k$ and $\mathbf{b}_k$ as

$$\text{cov}(\mathbf{a}_k, \mathbf{b}_k) := \left\langle \left( \mathbf{a}_k - \langle \mathbf{a}_k \rangle_k \right) \left( \mathbf{b}_k - \langle \mathbf{b}_k \rangle_k \right)^\top \right\rangle_k \tag{10}$$

and the limiting average is defined as

$$\langle \mathbf{a}_k \rangle_k = \lim_{K \to \infty} \frac{1}{K+1} \sum_{k=k_0}^{k_0+K} \mathbf{a}_k . \tag{11}$$

When possible, we will suppress $k$ and denote the average as $\langle \cdot \rangle$. The average is independent of the starting value $k_0$, therefore we can shift indices within the average, meaning $\langle \mathbf{a}_k \rangle = \langle \mathbf{a}_{k+l} \rangle$ for any $l \in \mathbb{Z}$.

To see this we take any integer $l \in \mathbb{Z}$ and instead of adding it to the index $k$ we can also subtract it from the starting value $k_0$ and then separate the sum into two sums,

$$\begin{aligned}
\langle \mathbf{a}_{k+l} \rangle &= \lim_{K \to \infty} \frac{1}{K+1} \sum_{k=k_0}^{k_0+K} \mathbf{a}_{k+l} \\
&= \lim_{K \to \infty} \frac{1}{K+1} \sum_{k=k_0-l}^{k_0-l+K} \mathbf{a}_k \\
&= \lim_{K \to \infty} \frac{1}{K+1} \sum_{k=k_0-l}^{k_0-1} \mathbf{a}_k + \lim_{K \to \infty} \frac{1}{K+1} \sum_{k=k_0}^{k_0-l+K} \mathbf{a}_k .
\end{aligned} \tag{12}$$

The first sum is independent of $K$ except for the factor $\frac{1}{K+1}$, so the limit of the first part is zero. The second part of the limit can be rearranged as follows,

$$\begin{aligned}
\lim_{K \to \infty} \frac{1}{K+1} \sum_{k=k_0}^{k_0-l+K} \mathbf{a}_k &= \lim_{K \to \infty} \frac{K-l+1}{K+1} \frac{1}{K-l+1} \sum_{k=k_0}^{k_0-l+K} \mathbf{a}_k \\
&= \lim_{\tilde{K} \to \infty} \frac{1}{\tilde{K}+1} \sum_{k=k_0}^{k_0+\tilde{K}} \mathbf{a}_k \tag{13} \\
&=: \langle \mathbf{a}_k \rangle , \tag{14}
\end{aligned}$$

where we used $\lim_{K \to \infty} \frac{K-l+1}{K+1} = 1$ and renamed $K - l$ in the second to last step. All together this gives us the desired relation $\langle \mathbf{a}_k \rangle = \langle \mathbf{a}_{k+l} \rangle$.

If one were to consider the covariance for a fixed update step $k$ averaged over multiple runs of SGD optimization, it is possible that this covariance could depend on the index $k$, but this is not our case of interest.

## B  VARIANCE CALCULATION

### B.1  COMMUTATIVITY OF THE COVARIANCE MATRICES

In this section, we show that if $[\mathbf{C}, \mathbf{H}] = 0$ also $\mathbf{\Sigma}$ and $\mathbf{\Sigma_v}$ will commute with $\mathbf{C}$, with $\mathbf{H}$ and with each other. We make the assumptions one to three from Section 4.3 and therefore the SGD update equations become

$$\mathbf{v}_k = -\eta \mathbf{H} \boldsymbol{\theta}_{k-1} + \beta \mathbf{v}_{k-1} - \eta \boldsymbol{\delta} \mathbf{g}_k , \tag{15}$$

$$\boldsymbol{\theta}_k = (\mathbf{1} - \eta \mathbf{H}) \boldsymbol{\theta}_{k-1} + \beta \mathbf{v}_{k-1} - \eta \boldsymbol{\delta} \mathbf{g}_k , \tag{16}$$

which can be rewritten by using the vector $\mathbf{y}_k := \begin{pmatrix} \boldsymbol{\theta}_k & \mathbf{v}_k \end{pmatrix}^\top$, combining both the current weight and velocity variable, to be

$$\mathbf{y}_{k+1} = \mathbf{X}\mathbf{y}_k - \mathbf{z}_{k+1} \ . \tag{17}$$

Here, $\mathbf{z}_k := \begin{pmatrix} \eta\boldsymbol{\delta}\mathbf{g}_k & \eta\boldsymbol{\delta}\mathbf{g}_k \end{pmatrix}^\top$ contains the current noise term, and the matrix governing the deterministic part of the update is defined to be

$$\mathbf{X} := \begin{pmatrix} \mathbf{1} - \eta\mathbf{H} & \beta\mathbf{1} \\ -\eta\mathbf{H} & \beta\mathbf{1} \end{pmatrix} \ . \tag{18}$$

By iteratively applying Equation (17) we obtain

$$\mathbf{y}_{k+h} = \mathbf{X}^h\mathbf{y}_k - \sum_{i=1}^{h} \mathbf{X}^{h-i}\mathbf{z}_{k+i} \ . \tag{19}$$

Under the assumption $0 \leq \beta < 1$ and $0 < \eta\lambda_i < 2(1 + \beta)$, for all eigenvalues $\lambda_i$ of $\mathbf{H}$, the magnitude of the eigenvalues of $\mathbf{X}$ will be less than one. It is straightforward to show this relation for the eigenvalues of $\mathbf{X}$ by using the eigenbasis of $\mathbf{H}$. Therefore,

$$\lim_{h \to \infty} \mathbf{X}^h\mathbf{y}_k = 0. \tag{20}$$

As we can shift the index in the weight variance, $h$ can be chosen arbitrarily large, which yields the following relation for the covariance

$$\langle \mathbf{y}_k\mathbf{y}_k^\top \rangle = \lim_{h \to \infty} \sum_{i,j=1}^{h} \mathbf{X}^{h-i}\langle \mathbf{z}_{k+i}\mathbf{z}_{k+j}^\top \rangle \left( \mathbf{X}^{h-j} \right)^\top \ . \tag{21}$$

Because Equation (19) together with Equation (20) implies $\langle \mathbf{y}_k \rangle = 0$ and therefore $\langle \boldsymbol{\theta}_k \rangle = 0$ and $\langle \mathbf{v}_k \rangle = 0$, the left hand side of Equation (21) contains the covariance matrices of interest,

$$\langle \mathbf{y}_k\mathbf{y}_k^\top \rangle = \begin{pmatrix} \boldsymbol{\Sigma} & \langle \boldsymbol{\theta}_k\mathbf{v}_k^\top \rangle \\ \langle \mathbf{v}_k\boldsymbol{\theta}_k^\top \rangle & \boldsymbol{\Sigma}_\mathbf{v} \end{pmatrix} \ . \tag{22}$$

From Equation (21) we can also infer that $\langle \mathbf{y}_k\mathbf{y}_k^\top \rangle$ is finite as the magnitude of the eigenvalues of $\mathbf{X}$ is less than one. Consequently, by Equation (22), the covariance matrices $\boldsymbol{\Sigma}$ and $\boldsymbol{\Sigma}_\mathbf{v}$ are finite as well. The average over the noise terms $\mathbf{z}_k$ on the right hand side of Equation (21) is by assumption equal to

$$\langle \mathbf{z}_{k+i}\mathbf{z}_{k+j}^\top \rangle = \eta^2 \left( \delta_{i,j} - \mathbf{1}_{\{1,\dots,M\}}(|i - j|)\frac{M - |i - j|}{M(M - 1)} \right) \cdot \begin{pmatrix} \mathbf{C} & \mathbf{C} \\ \mathbf{C} & \mathbf{C} \end{pmatrix} \ , \tag{23}$$

from which it follows that for any finite $h$ every matrix entry of the two by two super matrix on the right hand side of Equation (21) is a function of $\mathbf{C}$ and $\mathbf{H}$. Therefore, when considering the limit $h \to \infty$, $[\mathbf{C}, \mathbf{H}] = 0$ implies that $\boldsymbol{\Sigma}$ and $\boldsymbol{\Sigma}_\mathbf{v}$ will also commute with $\mathbf{C}$, with $\mathbf{H}$ and with each other.

### B.2 PROOF OF THE VARIANCE FORMULA FOR ONE SPECIFIC EIGENVALUE

Since $\boldsymbol{\Sigma}$ and $\boldsymbol{\Sigma}_\mathbf{v}$ will commute with $\mathbf{C}$, with $\mathbf{H}$ and with each other, it is sufficient to prove the one dimensional case. For the multidimensional case simply apply the proof in the direction of each common eigenvector individually. The expectation values discussed below are computed with respect to the asymptotic distributions of $\theta$ and $v$, since we are only interested in the asymptotic behavior of training. We want to find $\sigma_\theta^2 := \langle \theta_k\theta_k \rangle$ and $\sigma_v^2 := \langle v_kv_k \rangle$. We assume $0 \leq \beta < 1$ and $0 < \eta\lambda < 2(1 + \beta)$ where $\lambda$ is the Hessian eigenvalue.

The equations describing SGD in one dimension are:

$$g_k(\theta) = \frac{\partial}{\partial\theta}L(\theta) + \delta g_k(\theta) \tag{24}$$

$$v_k = -\eta g_k(\theta_{k-1}) + \beta v_{k-1} \tag{25}$$

$$\theta_k = \theta_{k-1} + v_k \ . \tag{26}$$

Our remaining assumptions can then be described the following way

$$L(\theta) = \frac{1}{2}\theta\lambda\theta \tag{27}$$

$$\delta g_k(\theta) = \delta g_k \tag{28}$$

$$\langle \delta g_k \delta g_{k+h} \rangle = \sigma_{\delta g}^2 \left( \delta_{h,0} - \mathbf{1}_{\{1,\dots,M\}}(|h|) \frac{M - |h|}{M(M-1)} \right) \tag{29}$$

$$\sigma_{\delta g}^2 := \langle \delta g_k \delta g_k \rangle . \tag{30}$$

With these assumptions the update equations can be described by a discrete stochastic linear equation of second order

$$\theta_k = (1 + \beta - \eta\lambda)\theta_{k-1} - \beta\theta_{k-2} - \eta\delta g_k \tag{31}$$

which can be rewritten into matrix form as follows

$$\mathbf{x}_k = \mathbf{D}\mathbf{x}_{k-1} - \eta\delta g_k \mathbf{e}_1 \tag{32}$$

$$\mathbf{x}_k := \begin{pmatrix} \theta_k \\ \theta_{k-1} \end{pmatrix} \tag{33}$$

$$\mathbf{e}_1 := \begin{pmatrix} 1 \\ 0 \end{pmatrix} \tag{34}$$

$$\mathbf{D} := \begin{pmatrix} 1 + \beta - \eta\lambda & -\beta \\ 1 & 0 \end{pmatrix} . \tag{35}$$

We are now interested in the following covariance matrix

$$\tilde{\mathbf{\Sigma}} := \langle \mathbf{x}_k \mathbf{x}_k^\top \rangle$$

$$= \begin{pmatrix} \sigma_\theta^2 & \langle \theta_k \theta_{k-1} \rangle \\ \langle \theta_k \theta_{k-1} \rangle & \sigma_\theta^2 \end{pmatrix} \tag{36}$$

where the second equality is due to the fact that $\langle \theta_k \theta_k \rangle = \langle \theta_{k-1} \theta_{k-1} \rangle$. As we are interested in the asymptotic covariance, this expectation value is independent of any finite shift of the index $k$. By inserting Equation (32) into $\langle \mathbf{x}_k \mathbf{x}_k^\top \rangle$ we arrive at the following equality

$$\langle \mathbf{x}_k \mathbf{x}_k^\top \rangle = \mathbf{D} \langle \mathbf{x}_{k-1} \mathbf{x}_{k-1}^\top \rangle \mathbf{D}^\top + \eta^2 \langle \delta g_k \delta g_k \rangle \mathbf{e}_1 \mathbf{e}_1^\top - \eta \left( \mathbf{D} \langle \mathbf{x}_{k-1} \delta g_k \rangle \mathbf{e}_1^\top + \left( \mathbf{D} \langle \mathbf{x}_{k-1} \delta g_k \rangle \mathbf{e}_1^\top \right)^\top \right) \tag{37}$$

which can be simplified to the equivalent equation

$$\tilde{\mathbf{\Sigma}} - \mathbf{D}\tilde{\mathbf{\Sigma}}\mathbf{D}^\top = \eta^2 \sigma_{\delta g}^2 \mathbf{e}_1 \mathbf{e}_1^\top - \eta \left( \mathbf{D} \langle \mathbf{x}_{k-1} \delta g_k \rangle \mathbf{e}_1^\top + \left( \mathbf{D} \langle \mathbf{x}_{k-1} \delta g_k \rangle \mathbf{e}_1^\top \right)^\top \right) . \tag{38}$$

If we apply the left-hand side on the vector $\mathbf{e}_1$, it can be expressed as

$$\left[ \tilde{\mathbf{\Sigma}} - \mathbf{D}\tilde{\mathbf{\Sigma}}\mathbf{D}^\top \right] \mathbf{e}_1 = \mathbf{F}_1^{-1} \tilde{\mathbf{\Sigma}} \mathbf{e}_1 \tag{39}$$

$$\mathbf{F}_1^{-1} := \begin{pmatrix} \eta\lambda(2 - \eta\lambda) - 2\beta(1 + \beta - \eta\lambda) & 2\beta(1 + \beta - \eta\lambda) \\ -(1 + \beta - \eta\lambda) & 1 + \beta \end{pmatrix} . \tag{40}$$

Also notice $v_k = \theta_k - \theta_{k-1}$ and therefore

$$\sigma_v^2 = 2\sigma_\theta^2 - 2\langle \theta_k \theta_{k-1} \rangle , \tag{41}$$

again due to the fact that the expectation value does not depend on k. Hence, the variances can then be expressed as

$$\begin{pmatrix} \sigma_\theta^2 \\ \sigma_v^2 \end{pmatrix} = \mathbf{F}_2 \tilde{\mathbf{\Sigma}} \mathbf{e}_1 \tag{42}$$

$$\mathbf{F}_2 := \begin{pmatrix} 1 & 0 \\ 2 & -2 \end{pmatrix} . \tag{43}$$

We define the matrix $\mathbf{F} := \mathbf{F}_2\mathbf{F}_1$. By applying both sides of Equation (38) to the vector $\mathbf{e}_1$, then multiplying by the matrix $\mathbf{F}$ from the left and using Equations (39) and (42) we obtain

$$\begin{pmatrix} \sigma_\theta^2 \\ \sigma_v^2 \end{pmatrix} = \mathbf{F}\left[\eta^2\sigma_{\delta g}^2\mathbf{e}_1\mathbf{e}_1^\top - \eta\left(\mathbf{D}\langle\mathbf{x}_{k-1}\delta g_k\rangle\,\mathbf{e}_1^\top + \left(\mathbf{D}\langle\mathbf{x}_{k-1}\delta g_k\rangle\,\mathbf{e}_1^\top\right)^\top\right)\right]\mathbf{e}_1 . \tag{44}$$

with

$$\mathbf{F} = \frac{1}{(1-\beta)\big(2(1+\beta)-\eta\lambda\big)}\begin{pmatrix} \frac{1+\beta}{\eta\lambda} & \frac{2\beta(\eta\lambda-1-\beta)}{\eta\lambda} \\ 2 & 2(\eta\lambda-2) \end{pmatrix} . \tag{45}$$

To simplify Equation (44) further we go back to Equation (32) and iterate it to obtain

$$\mathbf{x}_k = \mathbf{D}^n\mathbf{x}_{k-n} - \eta\sum_{h=0}^{n-1}\mathbf{D}^h\mathbf{e}_1\delta g_{k-h} . \tag{46}$$

We note that $\langle\mathbf{x}_{k-n}\delta g_k\rangle = 0$ for $n \geq M$. The correlation between noise terms separated by at least one epoch vanishes, and $\mathbf{x}_k$ only depends on past noise terms. By setting $n = M$ we find

$$\langle\mathbf{x}_{k-1}\delta g_k\rangle = \mathbf{D}^M\langle\mathbf{x}_{k-1-M}\delta g_k\rangle - \eta\sum_{h=0}^{M-1}\mathbf{D}^h\mathbf{e}_1\langle\delta g_k\delta g_{k-1-h}\rangle$$

$$= -\eta\sigma_{\delta g}^2\sum_{h=0}^{M-1}\mathbf{D}^h\mathbf{e}_1\left(-\frac{M-(h+1)}{M(M-1)}\right) , \tag{47}$$

where the assumption about the correlation of the noise terms, Equation (29), was inserted for the last line. Equation (47) is a sum of a finite geometric series and a derivative of that which can be simplified to

$$\langle\mathbf{x}_{k-1}\delta g_k\rangle = \eta\sigma_{\delta g}^2\frac{\mathbf{D}^M + (\mathbf{1}-\mathbf{D})M - \mathbf{1}}{(\mathbf{1}-\mathbf{D})^2M(M-1)}\mathbf{e}_1 . \tag{48}$$

Substituting this result back into Equation (44) yields

$$\begin{pmatrix} \sigma_\theta^2 \\ \sigma_v^2 \end{pmatrix} = \eta^2\sigma_{\delta g}^2\mathbf{F}\left[\mathbf{e}_1 - \left(\mathbf{E}+\mathbf{E}^\top\right)\mathbf{e}_1\right] \tag{49}$$

with the definition

$$\mathbf{E} := \mathbf{D}\frac{\mathbf{D}^M + (\mathbf{1}-\mathbf{D})M - \mathbf{1}}{(\mathbf{1}-\mathbf{D})^2M(M-1)}\mathbf{e}_1\mathbf{e}_1^\top . \tag{50}$$

With Equation (49) we have arrived at the exact formula for the variances which can easily be evaluated numerically.

### B.3 APPROXIMATION OF THE EXACT FORMULA

It is possible to approximate the exact result for the variance assuming small or large eigenvalues, respectively. For that, it is necessary to approximate $\mathbf{D}^M\mathbf{e}_1$. To do so, we will use the the following eigendecomposition of $\mathbf{D}$

$$\mathbf{D} = \mathbf{Q}\boldsymbol{\Lambda}\mathbf{Q}^{-1} \tag{51}$$

$$\boldsymbol{\Lambda} = \begin{pmatrix} \Lambda_+ & 0 \\ 0 & \Lambda_- \end{pmatrix} \tag{52}$$

$$\mathbf{Q} = \begin{pmatrix} \Lambda_+ & \Lambda_- \\ 1 & 1 \end{pmatrix} \tag{53}$$

$$\mathbf{Q}^{-1} = \frac{1}{\Lambda_+ - \Lambda_-}\begin{pmatrix} 1 & -\Lambda_- \\ -1 & \Lambda_+ \end{pmatrix} \tag{54}$$

$$\Lambda_\pm = \frac{1}{2}\left(1 + \beta - \eta\lambda \pm s\right) \tag{55}$$

$$s := \sqrt{(1-\beta)^2 - \eta\lambda\big(2(1+\beta)-\eta\lambda\big)} \tag{56}$$

It is straightforward to show that the magnitude of the eigenvalues of $\mathbf{D}$ is strictly smaller than one, $|\Lambda_\pm| < 1$, under the conditions $0 < \eta\lambda < 2(1+\beta)$ and $0 \leq \beta < 1$.

LARGE HESSIAN EIGENVALUES

$$\sigma_\theta^2 \approx \frac{\eta^2 \sigma_{\delta g}^2}{(1-\beta)(2(1+\beta)-\eta\lambda)} \cdot \frac{1+\beta}{\eta\lambda} \tag{57}$$

$$\sigma_v^2 \approx \frac{\eta^2 \sigma_{\delta g}^2}{(1-\beta)(2(1+\beta)-\eta\lambda)} \cdot 2 \tag{58}$$

We will show that this approximation for large Hessian eigenvalues is valid under the assumption $M(\eta\lambda)^2 \gg 1$ where $M$ is the number of batches per epoch. However, numerical studies indicate that these relations also hold under the previously mentioned assumptions of $\frac{M\eta\lambda}{1-\beta} \gg 1$, equivalent to $\lambda \gtrsim \lambda_{\text{cross}}$, and $M(1-\beta) \gg 1$.

Inserting the eigendecomposition of $\mathbf{D}$ into the expression $\mathbf{D}^M \mathbf{e_1}$ yields

$$\mathbf{D}^M \mathbf{e_1} = \begin{pmatrix} y_{M+1} \\ y_M \end{pmatrix} \tag{59}$$

$$y_M := \frac{\Lambda_+^M - \Lambda_-^M}{\Lambda_+ - \Lambda_-}. \tag{60}$$

From the definition of $y_M$ one sees that

$$y_M = \frac{\Lambda_+ + \Lambda_-}{2} y_{M-1} + \frac{\Lambda_+^{M-1} + \Lambda_-^{M-1}}{2} \,, \tag{61}$$

and by using $|\Lambda_\pm| < 1$ as well as $y_0 = 0$ one can show iteratively that

$$|y_M| \leq M + 1. \tag{62}$$

Therefore, we have

$$\left\| \mathbf{D}^M \mathbf{e_1} \right\|_\infty \leq M + 1 \tag{63}$$

where $\|\cdot\|_\infty$ is denoting the maximum norm $\|\mathbf{x}\|_\infty := \max_i |x_i|$ for a vector $\mathbf{x}$ or its induced matrix norm $\|\mathbf{A}\|_\infty := \max_i \sum_j |a_{ij}|$ for a matrix $\mathbf{A}$.

Explicit calculations show that

$$\left\| (\mathbf{1} - \mathbf{D})^{-1} \right\|_\infty \leq \frac{4}{\eta\lambda} \tag{64}$$

under the assumption that $0 \leq \beta < 1$ and $0 < \eta\lambda < 2(1+\beta)$. From here it is straightforward to show that

$$\left\| \left( \mathbf{E} + \mathbf{E}^\top \right) \mathbf{e_1} \right\|_\infty \leq \frac{\tilde{c}}{M(\eta\lambda)^2} \tag{65}$$

where $\tilde{c}$ is a factor of order unity under the constraints $0 \leq \beta < 1$ and $0 < \eta\lambda < 2(1+\beta)$. By substituting this result back into Equation (49) one directly sees that a comparison to the approximation yields

$$\left| 1 - \frac{\sigma_\theta^2}{\sigma_{\theta,\text{large}}^2} \right| \leq \frac{c_1}{M(\eta\lambda)^2} \,, \tag{66}$$

$$\left| 1 - \frac{\sigma_v^2}{\sigma_{v,\text{large}}^2} \right| \leq \frac{c_2}{M(\eta\lambda)^2} \,, \tag{67}$$

where $c_1$ and $c_2$ are again of order unity and the approximation is defined as

$$\begin{pmatrix} \sigma_{\theta,\text{large}}^2 \\ \sigma_{v,\text{large}}^2 \end{pmatrix} := \eta^2 \sigma_{\delta g}^2 \mathbf{F} \mathbf{e_1}$$

$$= \frac{\eta^2 \sigma_{\delta g}^2}{(1-\beta)(2(1+\beta)-\eta\lambda)} \cdot \begin{pmatrix} \frac{1+\beta}{\eta\lambda} \\ 2 \end{pmatrix}. \tag{68}$$

Interestingly, one can see that the approximation for large Hessian eigenvalues is equivalent to the result we would obtain if we assumed there was no autocorrelation of the noise to begin with.

In the case where the stricter assumption is not true, $M(\eta\lambda)^2 < 1$, but the numerically obtained conditions still hold, $\lambda \gtrsim \lambda_{\text{cross}}$ and $M(1-\beta) \gg 1$, it occurs that $\left\|\left(\mathbf{E} + \mathbf{E}^\top\right)\mathbf{e_1}\right\|_\infty$ is no longer small. But in that case, $\mathbf{F}\left(\mathbf{E} + \mathbf{E}^\top\right)\mathbf{e_1}$ can still be neglected compared to $\mathbf{Fe_1}$, as numerical experiments show.

SMALL HESSIAN EIGENVALUES

To obtain the relations for small Hessian eigenvalues, we perform a Taylor expansion with respect to $\lambda$ with the help of computer algebra. We neglect the terms which are at least of order $\lambda$. Numerical study indicates that these relations hold under the mentioned assumption of $\lambda \lesssim \lambda_{\text{cross}}$ and $M(1-\beta) \gg 1$.

It is straightforward but lengthy to obtain the following expression using the eigendecomposition of $\mathbf{D}$

$$\begin{pmatrix} \sigma_\theta^2 \\ \sigma_v^2 \end{pmatrix} = \frac{\eta^2 \sigma_{\delta g}^2}{2(1-\beta)(1+\beta)} \cdot \begin{pmatrix} \frac{M}{3}\frac{1+\beta}{1-\beta} + \mathcal{O}(\lambda) \\ 2 + \mathcal{O}(\lambda) \end{pmatrix} \tag{69}$$

where the zeroth order terms are simplified under approximation $M(1-\beta) \gg 1$.

### B.4 SATISFYING THE ASSUMPTIONS OF THE CORRELATION TIME RELATION

In this section we want to show that the weight and velocity variances resulting from stochastic gradient descent as described above and in Section 4.3 satisfies the necessary assumptions (i) to (iii) of Theorem 4.2 such that the velocity correlation time is equal to $\tau_i = 2\sigma_{\theta,i}^2/\sigma_{v,i}^2$. Validity of assumption (i) existence and finiteness of $\mathbf{\Sigma} := \text{cov}(\boldsymbol{\theta}_k, \boldsymbol{\theta}_k)$, $\mathbf{\Sigma_v} := \text{cov}(\mathbf{v}_k, \mathbf{v}_k)$, and $\langle\boldsymbol{\theta}\rangle$ can be inferred from the calculation presented in Appendix B.1. Therefore, we concentrate on assumption (ii) $\lim_{n\to\infty} \text{cov}(\boldsymbol{\theta}_k, \boldsymbol{\theta}_{k+n}) = 0$ and (iii) $\lim_{n\to\infty} n \cdot \text{cov}(\boldsymbol{\theta}_k, \boldsymbol{\theta}_{k+n} - \boldsymbol{\theta}_{k+n+1}) = 0$. We consider the one dimensional case, but the extension to the multidimensional case is straightforward. Additionally $\langle\boldsymbol{\theta}\rangle = 0$ (see Appendix B.1) and, therefore, the remaining two assumptions (ii) and (iii) can be written as $\lim_{m\to\infty} \langle\theta_k\theta_{k+m}\rangle = 0$ and $\lim_{m\to\infty} m\left(\langle\theta_k\theta_{k+m}\rangle - \langle\theta_k\theta_{k+m+1}\rangle\right) = 0$.

We will now show that for stochastic gradient descent under the assumptions of Section 4.3 the more restrictive relation $\lim_{m\to\infty} m\langle\theta_k\theta_{k+m}\rangle = 0$ is satisfied, from which follows (ii) and (iii). Following Appendix B.2 and using the same notation, we have the relation

$$\langle\mathbf{x}_k\mathbf{x}_{k-m}^\top\rangle = \mathbf{D}\langle\mathbf{x}_{k-1}\mathbf{x}_{k-m}^\top\rangle - \eta\mathbf{e_1}\langle\delta g_k\mathbf{x}_{k-m}^\top\rangle , \tag{70}$$

where $\mathbf{x}_k := \begin{pmatrix} \theta_k & \theta_{k-1} \end{pmatrix}^\top$. For $m > M$, with $M$ being the number of batches per epoch, the correlation with the noise term on the right hand side of Equation (70) is equal to zero as discussed in Appendix B.2. By iterating Equation (70), for $m > M$ we have

$$\langle\mathbf{x}_k\mathbf{x}_{k-m}^\top\rangle = \mathbf{D}^{m-M-1}\langle\mathbf{x}_{k-m+M+1}\mathbf{x}_{k-m}^\top\rangle$$
$$= \mathbf{D}^{m-M-1}\langle\mathbf{x}_k\mathbf{x}_{k-M-1}^\top\rangle . \tag{71}$$

As described in Appendix B.2, the magnitude of both eigenvalues of $\mathbf{D}$ is strictly smaller than one. This implies that there exists a matrix norm $\|\cdot\|_{\mathbf{D}}$ such that $\|\mathbf{D}\|_{\mathbf{D}} < 1$ from which one can deduce

$$\left\|\langle\mathbf{x}_k\mathbf{x}_{k-m}^\top\rangle\right\|_{\mathbf{D}} \leq \|\mathbf{D}\|_{\mathbf{D}}^{m-M-1} \cdot \left\|\langle\mathbf{x}_k\mathbf{x}_{k-M-1}^\top\rangle\right\|_{\mathbf{D}} . \tag{72}$$

Taking the limit of $m \to \infty$ we obtain

$$\lim_{m\to\infty} m\left\|\langle\mathbf{x}_k\mathbf{x}_{k-m}^\top\rangle\right\|_{\mathbf{D}} \leq \text{const} \cdot \lim_{m\to\infty} m\|\mathbf{D}\|_{\mathbf{D}}^{m-M-1}$$
$$= 0 , \tag{73}$$

and because

$$\langle\mathbf{x}_k\mathbf{x}_{k-m}^\top\rangle = \begin{pmatrix} \langle\theta_k\theta_{k-m}\rangle & \langle\theta_k\theta_{k-m-1}\rangle \\ \langle\theta_{k-1}\theta_{k-m}\rangle & \langle\theta_{k-1}\theta_{k-m-1}\rangle \end{pmatrix} \tag{74}$$

we finally find

$$\lim_{m\to\infty} m\langle\theta_k\theta_{k-m}\rangle = 0$$

$$\Rightarrow \lim_{m\to\infty} m\langle\theta_k\theta_{k+m}\rangle = 0 \ . \tag{75}$$

## C  CALCULATION OF THE CORRELATION TIME RELATION

We want to prove Theorem 4.2, that is, the relation

$$\frac{2\sigma_{\theta,i}^2}{\sigma_{v,i}^2} = \frac{\sum_{n=1}^\infty n\,\langle v_{k,i}v_{k+n,i}\rangle}{\sum_{n=1}^\infty \langle v_{k,i}v_{k+n,i}\rangle} \ , \tag{76}$$

under the following three assumptions: (i) Existence and finiteness of $\mathbf{\Sigma} := \mathrm{cov}(\boldsymbol{\theta}_k,\boldsymbol{\theta}_k)$, $\mathbf{\Sigma_v} := \mathrm{cov}(\mathbf{v}_k,\mathbf{v}_k)$, and $\langle\boldsymbol{\theta}\rangle$. (ii) $\lim_{n\to\infty}\mathrm{cov}(\boldsymbol{\theta}_k,\boldsymbol{\theta}_{k+n}) = 0$. (iii) $\lim_{n\to\infty} n\cdot\mathrm{cov}(\boldsymbol{\theta}_k,\boldsymbol{\theta}_{k+n}-\boldsymbol{\theta}_{k+n+1}) = 0$. For example, the latter two assumptions hold true if the weight correlation function decays as $\mathrm{cov}(\boldsymbol{\theta}_k,\boldsymbol{\theta}_{k+n}) \propto n^{-2}$ or faster. In the setup described in Section 4.3, the weight correlations will even decay exponentially fast (see Appendix B.4).

We assume that that $\langle\boldsymbol{\theta}\rangle$, $\mathbf{\Sigma_\theta}$ and $\mathbf{\Sigma_v}$ exist and are finite. Without loss of generality, let $\langle\boldsymbol{\theta}\rangle = 0$. We consider only the one-dimensional case. For the multidimensional case, simply apply the proof in the direction of any basis vector individually. Note, that the relation still holds if $[\mathbf{\Sigma},\mathbf{\Sigma_v}] \neq 0$. In this case, $\sigma_{\theta,i}^2$ and $\sigma_{v,i}^2$ would just be the variances of the weight and the velocity in the given direction but no longer necessarily eigenvalues of $\mathbf{\Sigma}$ and $\mathbf{\Sigma_v}$.

The remaining two assumptions (ii) and (iii) of Theorem 4.2 can now be written as

$$\lim_{m\to\infty} \langle\theta_k\theta_{k+m}\rangle = 0 \tag{77}$$

$$\lim_{m\to\infty} m\big(\langle\theta_k\theta_{k+m}\rangle - \langle\theta_k\theta_{k+m+1}\rangle\big) = 0 \ . \tag{78}$$

We begin the proof with the following chain of equations

$$\sigma_\theta^2 = \langle\theta_k^2\rangle$$
$$= \left\langle(\theta_k - \theta_{k+J} + \theta_{k+J})^2\right\rangle$$
$$= \left\langle(\theta_k - \theta_{k+J})^2\right\rangle - 2\left\langle\theta_{k+J}^2\right\rangle + 2\left\langle\theta_k\theta_{k+J}\right\rangle + \left\langle\theta_{k+J}^2\right\rangle \ , \tag{79}$$

which holds for any integer $J$. We have $\langle\theta_{k+J}^2\rangle = \langle\theta_k^2\rangle$ since the expectation value cannot depend on $k$. Additionally, by definition we have $v_k = \theta_k - \theta_{k-1}$ which yields

$$\theta_k - \theta_{k+J} = \sum_{i=1}^J v_{k+i} \ . \tag{80}$$

Therefore, we can rewrite Equation (79) as follows

$$2\sigma_\theta^2 = 2\langle\theta_k\theta_{k+J}\rangle + \sum_{i,j=1}^J \langle v_{k+i}v_{k+j}\rangle$$

$$= 2\langle\theta_k\theta_{k+J}\rangle + \sum_{i,j=1}^J \langle v_k v_{k+j-i}\rangle$$

$$= 2\langle\theta_k\theta_{k+J}\rangle + \sum_{m=0}^{J-1}\sum_{n=-m}^m \langle v_k v_{k+n}\rangle \ , \tag{81}$$

where we first shifted the index within the expectation value and then restructured the sum by defining $m := \max(i,j) - 1$ and $n := j - i$. We now take the limit of $J \to \infty$ and because of

Equation (77) and the assumption of a finite $\sigma_\theta^2$ we have

$$\sum_{m=0}^\infty \sum_{n=-m}^m \langle v_k v_{k+n} \rangle < \infty \tag{82}$$

$$\Rightarrow \sum_{n=-\infty}^\infty \langle v_k v_{k+n} \rangle = 0 \ . \tag{83}$$

We note that $\langle v_k v_{k+n} \rangle = \langle v_k v_{k-n} \rangle$ because we can shift the index, and the two factors commute. Substituting this relation into Equation (83) yields

$$\sum_{n=1}^\infty \langle v_k v_{k+n} \rangle = -\frac{1}{2} \langle v_k v_k \rangle$$

$$= -\frac{1}{2} \sigma_v^2 \ . \tag{84}$$

For the second part of the proof we will start again with $v_k = \theta_k - \theta_{k-1}$ and the following sum

$$\sum_{n=1}^m n \langle v_k v_{k+n} \rangle = \sum_{n=1}^m n \big( 2\langle \theta_k \theta_{k+n} \rangle - \langle \theta_{k-1} \theta_{k+n} \rangle - \langle \theta_k \theta_{k+n-1} \rangle \big)$$

$$= -\langle \theta_k \theta_k \rangle + \langle \theta_k \theta_{k+m} \rangle + m\big( \langle \theta_k \theta_{k+m} \rangle - \langle \theta_k \theta_{k+m+1} \rangle \big) \ , \tag{85}$$

where nearly all terms cancel each other again due to the fact that we can shift the index within the expectation value. By taking the limit $m \to \infty$ and using the assumptions (ii) and (iii) (Equations (77) and (78)) we have

$$\sum_{n=1}^\infty n \langle v_k v_{k+n} \rangle = -\langle \theta_k \theta_k \rangle \ . \tag{86}$$

Finally, by dividing Equation (86) by Equation (84) we arrive at the final expression

$$\frac{2\sigma_{\theta,i}^2}{\sigma_{v,i}^2} = \frac{\sum_{n=1}^\infty n \langle v_k v_{k+n} \rangle}{\sum_{n=1}^\infty \langle v_k v_{k+n} \rangle} \ . \tag{87}$$

## D  CALCULATION OF THE NOISE AUTOCORRELATION

We want to calculate the autocorrelation function of epoch-based SGD for a fixed weight vector $\boldsymbol{\theta}$ and under the assumption that the total number of examples is an integer multiple of the number of examples per batch. For that we repeat the following definitions:

$$\boldsymbol{\delta}\mathbf{g}_k(\boldsymbol{\theta}) := \frac{1}{S} \sum_{n \in \mathcal{B}_k} \boldsymbol{\nabla}\big( l(\boldsymbol{\theta}, x_n) - L(\boldsymbol{\theta}) \big) \tag{88}$$

$$\mathcal{B}_k = \{n_1, ..., n_S\} \dots \text{ batch of step } k, \text{ sampling without replacement within epoch} \tag{89}$$

$$n_j \in \{1, \dots, N\} \tag{90}$$

$$N \dots \text{ total number of examples} \tag{91}$$

$$S \dots \text{ number of examples per batch} \tag{92}$$

We can rewrite the noise terms as follows:

$$\boldsymbol{\delta}\mathbf{g}_k(\boldsymbol{\theta}) = \frac{1}{S} \sum_{n \in \mathcal{B}_k} \boldsymbol{\delta}\mathbf{g}_\mathbf{e}(n, \boldsymbol{\theta})$$

$$= \frac{1}{S} \sum_{n=1}^N \boldsymbol{\delta}\mathbf{g}_\mathbf{e}(n, \boldsymbol{\theta}) s_k^n \tag{93}$$

$$s_k^n := \mathbf{1}_{\mathcal{B}_k}(n)$$

$$= \begin{cases} 1 \text{ if } n \in \mathcal{B}_k \\ 0 \text{ if } n \notin \mathcal{B}_k \end{cases} \tag{94}$$

$$\boldsymbol{\delta}\mathbf{g}_\mathbf{e}(n, \boldsymbol{\theta}) := \boldsymbol{\nabla}\big( l(\boldsymbol{\theta}, x_n) - L(\boldsymbol{\theta}) \big) \ . \tag{95}$$

Let $h \geq 0$ be fixed. The correlation matrix can be expressed as

$$\text{cov}\big(\delta \mathbf{g}_k(\boldsymbol{\theta}), \delta \mathbf{g}_{k+h}(\boldsymbol{\theta})\big) = \mathbb{E}\left[\delta \mathbf{g}_k(\boldsymbol{\theta}) \delta \mathbf{g}_{k+h}(\boldsymbol{\theta})^\top\right]$$

$$= \frac{1}{S^2} \sum_{n,\tilde{n}=1}^{N} \delta \mathbf{g}_{\mathbf{e}}(n, \boldsymbol{\theta}) \delta \mathbf{g}_{\mathbf{e}}(\tilde{n}, \boldsymbol{\theta})^\top \mathbb{E}\left[s_k^n s_{k+h}^{\tilde{n}}\right] . \tag{96}$$

The expectation value of $s_k^n = \mathbf{1}_{\mathcal{B}_k}(n)$ is the probability that example $n$ is part of batch $k$. Because every example is equally likely to appear in a given batch, this probability is equal to $S/N$.

$$\mathbb{E}\left[s_k^n\right] = \text{P}(s_k^n = 1)$$

$$= \frac{S}{N} . \tag{97}$$

Similarly we can calculate the desired correlation:

$$\mathbb{E}\left[s_k^n s_{k+h}^{\tilde{n}}\right] = \text{P}\big(s_k^n = 1, s_{k+h}^{\tilde{n}} = 1\big)$$

$$= \text{P}(s_k^n = 1)\,\text{P}\big(s_{k+h}^{\tilde{n}} = 1 \mid s_k^n = 1\big)$$

$$= \frac{S}{N}\,\text{P}\big(s_{k+h}^{\tilde{n}} = 1 \mid s_k^n = 1\big) . \tag{98}$$

The last term can be split up into different probabilities for different values of h. We can also distinguish the case where the two steps $k$ and $k+h$ are within the same epoch ($\text{ep}(k) = \text{ep}(k+h)$) or in different epochs ($\text{ep}(k) \neq \text{ep}(k+h)$).

$$\text{P}\big(s_{k+h}^{\tilde{n}} = 1 \mid s_k^n = 1\big) = \delta_{h,0} \cdot \text{P}\big(s_k^{\tilde{n}} = 1 \mid s_k^n = 1\big) + (1 - \delta_{h,0}) \cdot$$

$$\Big[\text{P}\big(\text{ep}(k) = \text{ep}(k+h)\big)\text{P}\big(s_{k+h}^{\tilde{n}} = 1 \mid s_k^n = 1, \text{ep}(k) = \text{ep}(k+h)\big) +$$

$$\text{P}\big(\text{ep}(k) \neq \text{ep}(k+h)\big)\text{P}\big(s_{k+h}^{\tilde{n}} = 1 \mid s_k^n = 1, \text{ep}(k) \neq \text{ep}(k+h)\big)\Big] . \tag{99}$$

The first term of the right hand side of Equation (99) is the probability that a given example occurs in a batch, assuming that we already know one of the examples of that batch.

$$\text{P}\big(s_k^{\tilde{n}} = 1 \mid s_k^n = 1\big) = \text{P}(s_k^{\tilde{n}} = 1 \mid s_k^n = 1) \cdot \delta_{n,\tilde{n}} + \text{P}\big(s_k^{\tilde{n}} = 1 \mid s_k^n = 1, n \neq \tilde{n}\big)\,(1 - \delta_{n,\tilde{n}})$$

$$= 1 \cdot \delta_{n,\tilde{n}} + \frac{S-1}{N-1}\,(1 - \delta_{n,\tilde{n}})$$

$$= \frac{N-S}{N-1}\,\delta_{n,\tilde{n}} + \text{const.} \tag{100}$$

The second term of Equation (99) is multiplied by $(1 - \delta_{h,0})$. Therefore, we assume $h \geq 1$ for the following argument. That is, we want to know the probabilities under the assumption that we are comparing examples from different batches. If the two batches are still from the same epoch, examples cannot repeat as the total number of examples is an integer multiple of the number of examples per batch and because of that every example is shown only once per epoch. Therefore, for $h \geq 1$ holds:

$$\text{P}\big(s_{k+h}^{\tilde{n}} = 1 \mid s_k^n = 1, \text{ep}(k) = \text{ep}(k+h)\big) = 0 \cdot \delta_{n,\tilde{n}} + \frac{S}{N-1}\,(1 - \delta_{n,\tilde{n}})$$

$$= -\frac{S}{N-1}\delta_{n,\tilde{n}} + \text{const.} \tag{101}$$

If we consider batches from different epochs, the probability becomes independent of the given examples:

$$\text{P}\big(s_{k+h}^{\tilde{n}} = 1 \mid s_k^n = 1, \text{ep}(k) \neq \text{ep}(k+h)\big) = \frac{S}{N}$$

$$= \text{const.} \tag{102}$$

Lastly, we need to know the probability that two given batches $k$ and $k+h$ are from the same epoch:

$$\mathrm{P}\big(\mathrm{ep}(k) = \mathrm{ep}(k+h)\big) = \mathbf{1}_{\{1,\ldots,M\}}(h)\,\frac{M-h}{M}\;, \tag{103}$$

where $M = N/S$ is again the number of batches per epoch.

We can now combine all derived probabilities and arrive at the following relation:

$$\mathbb{E}\left[s_k^n s_{k+h}^{\tilde{n}}\right] = \delta_{n,\tilde{n}}\,\frac{S}{N}\frac{N-S}{N-1}\left(\delta_{h,0} - \mathbf{1}_{\{1,\ldots,M\}}(h)\frac{S}{N-S}\frac{M-h}{M}\right) + \mathrm{const.}$$

$$= \delta_{n,\tilde{n}}\,S^2\left(\frac{1}{S} - \frac{1}{N}\right)\frac{1}{N-1}\left(\delta_{h,0} - \mathbf{1}_{\{1,\ldots,M\}}(h)\frac{M-h}{M(M-1)}\right) + \mathrm{const.} \tag{104}$$

If we now also consider negative values for h, the expression depends only on the absolute value of h due to symmetry.

By using the following two helpful relations:

$$\sum_{n,\tilde{n}=1}^{N} \boldsymbol{\delta}\mathbf{g_e}(n,\boldsymbol{\theta})\boldsymbol{\delta}\mathbf{g_e}(\tilde{n},\boldsymbol{\theta})^\top \delta_{n,\tilde{n}} = \sum_{n=1}^{N} \boldsymbol{\delta}\mathbf{g_e}(n,\boldsymbol{\theta})\boldsymbol{\delta}\mathbf{g_e}(n,\boldsymbol{\theta})^\top$$

$$=: (N-1)\,\mathbf{C}_0(\boldsymbol{\theta})\;, \tag{105}$$

$$\sum_{n,\tilde{n}=1}^{N} \boldsymbol{\delta}\mathbf{g_e}(n,\boldsymbol{\theta})\boldsymbol{\delta}\mathbf{g_e}(\tilde{n},\boldsymbol{\theta})^\top \cdot 1 = \left(\sum_{n=1}^{N} \boldsymbol{\delta}\mathbf{g_e}(n,\boldsymbol{\theta})\right)\left(\sum_{n=1}^{N} \boldsymbol{\delta}\mathbf{g_e}(n,\boldsymbol{\theta})^\top\right)$$

$$= \boldsymbol{\nabla}\big(L(\boldsymbol{\theta}) - L(\boldsymbol{\theta})\big)\boldsymbol{\nabla}^\top\big(L(\boldsymbol{\theta}) - L(\boldsymbol{\theta})\big)$$

$$= 0\;, \tag{106}$$

we can insert the expectation value $\mathbb{E}\left[s_k^n s_{k+h}^{\tilde{n}}\right]$ into Equation (96) and arrive at the final expression:

$$\mathrm{cov}\big[\boldsymbol{\delta}\mathbf{g}_k(\boldsymbol{\theta}), \boldsymbol{\delta}\mathbf{g}_{k+h}(\boldsymbol{\theta})\big] = \mathrm{cov}\big[\boldsymbol{\delta}\mathbf{g}_k(\boldsymbol{\theta}), \boldsymbol{\delta}\mathbf{g}_k(\boldsymbol{\theta})\big] \cdot \left(\delta_{h,0} - \mathbf{1}_{\{1,\ldots,M\}}(|h|)\frac{M-|h|}{M(M-1)}\right), \tag{107}$$

$$\mathrm{cov}\big[\boldsymbol{\delta}\mathbf{g}_k(\boldsymbol{\theta}), \boldsymbol{\delta}\mathbf{g}_k(\boldsymbol{\theta})\big] = \left(\frac{1}{S} - \frac{1}{N}\right)\mathbf{C}_0(\boldsymbol{\theta})\;. \tag{108}$$

# E    COMPARISON WITH PRINCIPAL COMPONENT ANALYSIS

Our approach to analysis sets itself apart from that of Feng & Tu (2021) principally in the selection of the basis $\{\mathbf{p_i},\, i = 1, \ldots, d\}$ used for examining the weights. While they employ the principal components of the weight series - the eigenvectors of $\boldsymbol{\Sigma}$ - we use the eigenvectors of the Hessian matrix $\mathbf{H}(\boldsymbol{\theta_K})$ computed at the beginning of the analysis period.

This choice enables us to directly create plots of variances and correlation time against the Hessian eigenvalue for each corresponding direction. Feng & Tu devised a landscape-dependent flatness parameter $F_i$ for every direction $\mathbf{p_i}$. However, with the assistance of the second derivative $F_i \approx \big(\partial^2 L(\boldsymbol{\theta})/\partial\theta_i^2\big)^{-\frac{1}{2}}$, where $\theta_i = \boldsymbol{\theta}\cdot\mathbf{p_i}$, this parameter can be approximated, provided this second derivative retains a sufficiently positive value. Hence, in the eigenbasis of the Hessian matrix, the flatness parameter can be approximated as $F_i \approx \lambda_i^{-\frac{1}{2}}$, facilitating comparability between our analysis and that of Feng & Tu.

The principal component basis, as used by Feng & Tu, holds a distinct advantage. For our analysis, we needed to eliminate the near-linear trajectory of the weights by deducting the mean velocity. However, in Feng & Tu's analysis, this movement is automatically subsumed in the first principal component due to its pronounced variance. Hence, there's no necessity for additional subtraction of this drift in the weight covariance eigenbasis.

Yet, the weight covariance eigenbasis has a significant shortcoming: it yields artifacts. This is because $\boldsymbol{\Sigma}$ is calculated as an average over a finite data set, skewing its eigenvalues from the anticipated distribution. Consequently, the resultant eigenvectors may not align perfectly with the

expected ones. This issue is further exacerbated due to the high dimensionality of the underlying space.

The artifact issue becomes evident in Figure 3, which displays synthetic data generated through stochastic gradient descent within an isotropic quadratic potential coupled with isotropic noise. With 2,500 dimensions, the model mirrors the scale of a layer in the fully connected neural network that Feng & Tu investigated. The weight series comprises 12,000 steps, which correspond to ten epochs of training this network. Analyzing this data with the weight covariance eigenbasis seemingly suggests anisotropic variance and correlation time. However, if the data is inspected without any basis change, both the variance and correlation time appear isotropically distributed as anticipated.

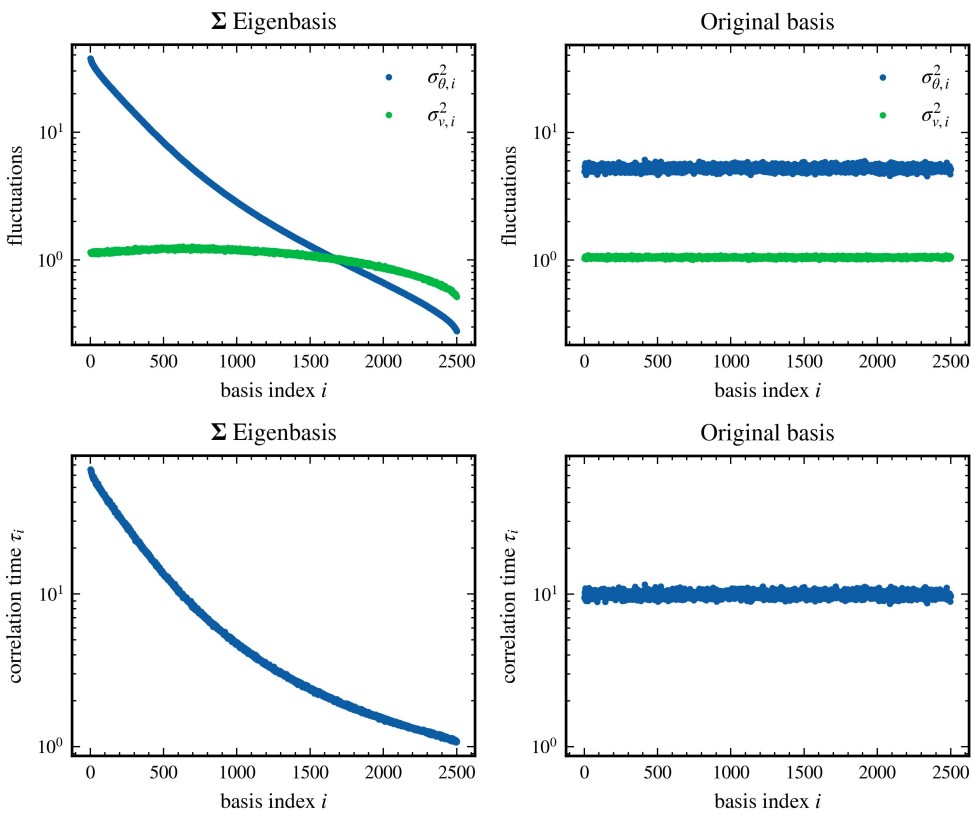

Figure 3: Comparison of weight and velocity fluctuations for synthetic data analyzed in two different bases. We define the variance in weight, $\sigma_{\theta,i}^2$, as $\mathbf{p_i}^\top \mathbf{\Sigma} \mathbf{p_i}$, and the variance in velocity, $\sigma_{v,i}^2$, as $\mathbf{p_i}^\top \mathbf{\Sigma_v} \mathbf{p_i}$. The correlation time, $\tau_i$, is given by $2\sigma_{\theta,i}^2/\sigma_{v,i}^2$. The synthetic data was generated by simulating SGD for 12,000 steps within a 2,500-dimensional space featuring an isotropic quadratic potential and isotropic noise. In the original basis analysis, both the variance and correlation time, as expected, retain isotropy. However, when the analysis is conducted in the eigenbasis of the weight covariance matrix, a pronounced anisotropy emerges.

To navigate around this key issue associated with the eigenbasis of $\mathbf{\Sigma}$, we adopted the eigenvectors of the Hessian matrix. Unlike $\mathbf{\Sigma}$, the Hessian is not computed as an average over update steps but can, in theory, be precisely calculated for any given weight vector. Consequently, the Hessian matrix does not suffer from finite size effects. The difference between these two bases for actual data is visible in Figure 4. Here, we analyzed only the weights of the first convolutional layer of the LeNet from the main text to ensure comparability with Feng & Tu's results. In this specific comparison, the network was trained without weight decay. Due to this and the fact that we are only investigating the weights of one layer, $\lambda_{\mathrm{cross}}$ is significantly larger than all Hessian eigenvalues. As a result, when analyzing in the eigenbasis of the Hessian matrix related to this layer, both the variance and the correlation time align well with the prediction for smaller Hessian eigenvalues.

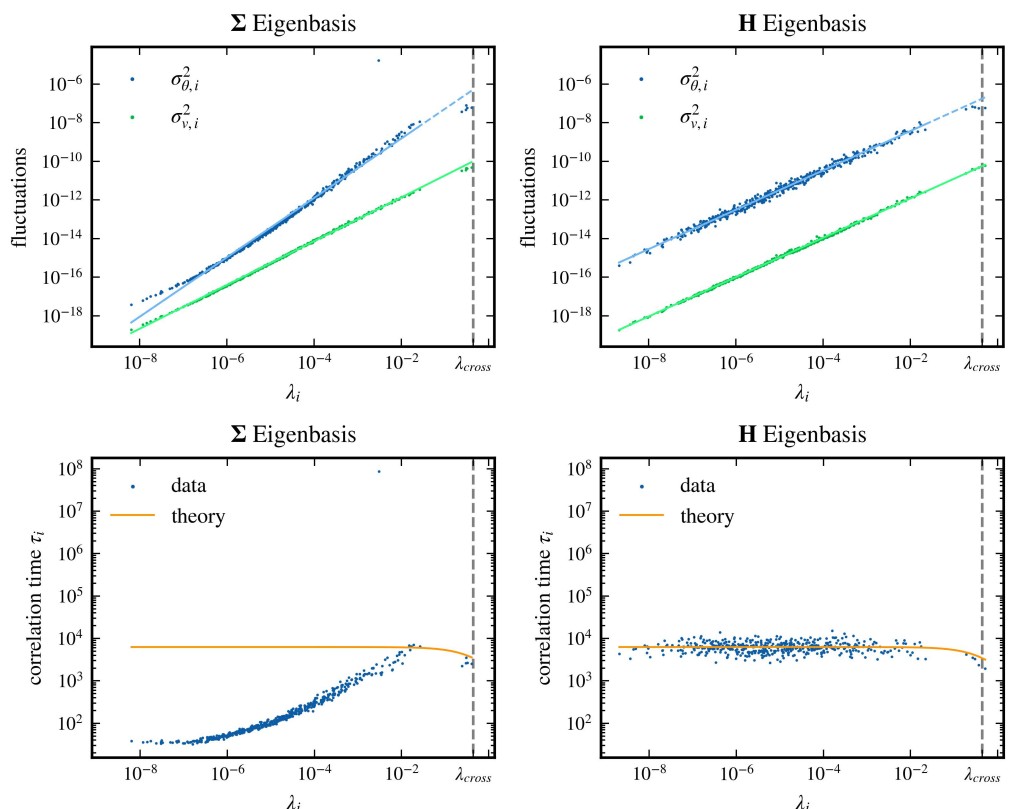

Figure 4: Comparison of weight and velocity variances for all 450 weights of the first convolutional layer of the LeNet, as discussed in the main text, analyzed in two different bases. In order to facilitate a more directly comparable analysis to Feng & Tu (2021), the network was trained without weight decay for this specific analysis and the analysis period was limited to 10 epochs, as opposed to the usual 20 epochs. The columns represent different bases: for the left column $\mathbf{p_i}$ are the eigenvectors of $\mathbf{\Sigma}$ and for the right column $\mathbf{p_i}$ are the eigenvectors of $\mathbf{H}$. The mean velocity was subtracted in the right column. The rows illustrate the weight and velocity variance, $\sigma_{\theta,i}^2 = \mathbf{p_i}^\top \mathbf{\Sigma} \mathbf{p_i}$, $\sigma_{v,i}^2 = \mathbf{p_i}^\top \mathbf{\Sigma_v} \mathbf{p_i}$ (top row), and the correlation time $\tau_i = 2\sigma_{\theta,i}^2/\sigma_{v,i}^2$ (bottom row). The second derivative of the corresponding direction is depicted on the x-axis, $\lambda_i = \mathbf{p_i}^\top \mathbf{H} \mathbf{p_i}$. The top row solid lines indicate fit regions for a linear fit. For the $\mathbf{H}$ eigenbasis, the respective exponent of the power law relation is $1.018 \pm 0.008$ for weight variance and $1.017 \pm 0.002$ for velocity variance with a $2\sigma$-error. For the $\mathbf{\Sigma}$ eigenbasis, the corresponding exponent is $1.537 \pm 0.012$ for weight variance and $1.134 \pm 0.002$ for velocity variance.

However, analyzing in the eigenbasis of the weight covariance matrix, the correlation time appears heavily dependent on the second derivative of the loss in the given direction. Additionally, the relationship between the weight variance and the second derivative shifts and more closely aligns with Feng & Tu's results as the power law exponent is significantly larger than one. The first principal component, which Feng & Tu referred to as the drift mode, stands out due to its unusually long correlation time. This is to be expected, as this is the direction in which the weights are moving at an approximately constant velocity.

# F  HESSIAN EIGENVALUE DENSITY

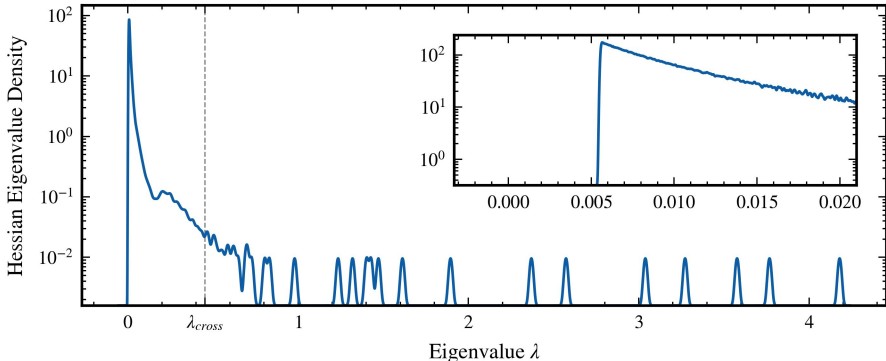

Figure 5: The distribution of the approximated 5,000 Hessian eigenvalues of the LeNet discussed in the main text. The inset shows that the smallest approximated eigenvalue has a magnitude of about 0.005.

# G  DRAWING WITH REPLACEMENT

To confirm that the results obtained are indeed affected by the correlations present in SGD noise, due to the epoch-based learning strategy, we reapply the analysis described in the main text. In this instance, however, we deviate from our previous method of choosing examples for each batch within an epoch without replacement. Instead, we select examples with replacement from the complete pool of examples for every batch. This modification during the analysis period allows a more complete assessment of the impact of correlations on the derived results.

Figure 6 offers clear visual proof that when examples are selected with replacement, the previously noted anti-correlations within the SGD noise vanish. This observation confirms our hypothesis that the anti-correlations mentioned in the main text are indeed an outcome of the epoch-based learning technique. Consequently, we can predict that this change will influence the behaviour of the weight and velocity variance. As previously discussed, the theoretical results we have achieved for Hessian eigenvectors with eigenvalues exceeding $\lambda_{\text{cross}}$ conform to what one would predict in the absence of any correlation within the noise. Therefore, when examples are drawn with replacement, we anticipate the weight variance to be isotropic in all directions, while the velocity variance should remain unchanged.

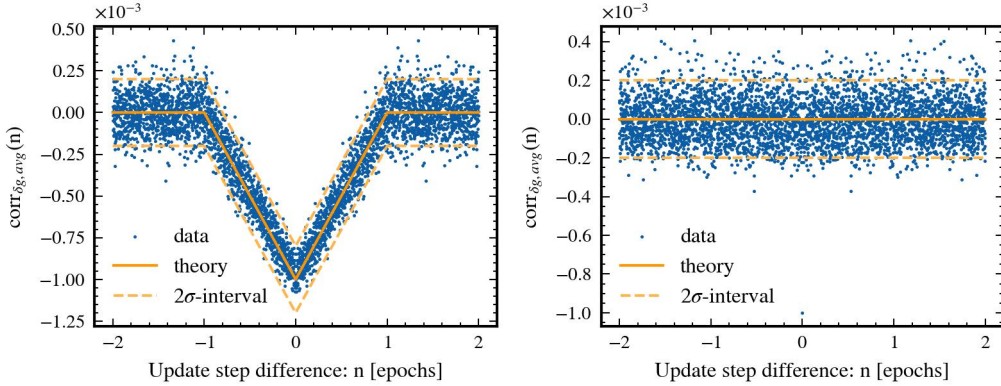

Figure 6: Autocorrelations of the SGD noise compared for drawing examples without replacement (left) and with replacement (right).

Upon reviewing Figure 7, it is clear that the velocity variance stays unchanged as predicted. However, while the weight variance remains constant for a broader subset of Hessian eigenvalues, it reduces for extremely small eigenvalues. Likewise, the correlation time is still limited for these minuscule Hessian eigenvalues. These deviations can be attributed to the finite time frame of the analysis period, comprising 20,000 update steps. This limited time window sets a cap on the maximum correlation time, consequently leading to a decreased weight variance for these small Hessian eigenvalues. Despite this, it is noteworthy that this maximum correlation time is still roughly one order of magnitude longer than the maximum correlation time induced by the correlations arising from the epoch-based learning approach.

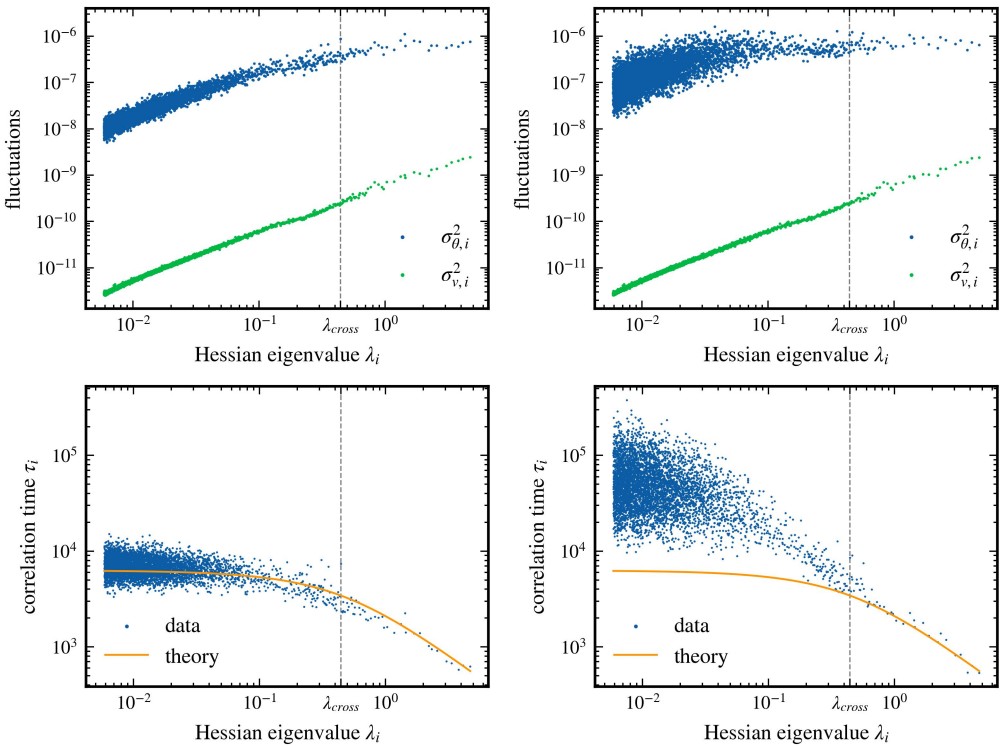

Figure 7: Relationship between Hessian eigenvalues and the variances of weights and velocities, as well as correlation times. For the left column the examples are drawn in epochs without replacement and for the right column the examples are drawn with replacement.

# H   TRAINING SCHEDULE OF LENET

In order to corroborate our theoretical findings, we have conducted a small-scale experiment. We have trained a LeNet architecture, similar to the one described in (Feng & Tu, 2021), using the CIFAR10 dataset (Krizhevsky, 2009). LeNet is a compact convolutional network comprised of two convolutional layers followed by three dense layers. The network comprises approximately 137,000 parameters. As our loss function, we employed Cross Entropy, along with an L2 regularization with a prefactor of $10^{-4}$. We used SGD to train the network for 100 epochs, employing an exponential learning rate schedule that reduces the learning rate by a factor of $0.98$ each epoch. The initial learning rate is set at $5 \cdot 10^{-3}$, which eventually reduces to approximately $7 \cdot 10^{-4}$ after 100 epochs. The momentum parameter and the minibatch size $S$ are set to $0.9$ and $50$, respectively, which results in a thousand minibatches per epoch, $M = 1000$. This setup achieves 100% training accuracy and 63% testing accuracy. The evolution of loss and accuracy during training can be seen in Figure 8.

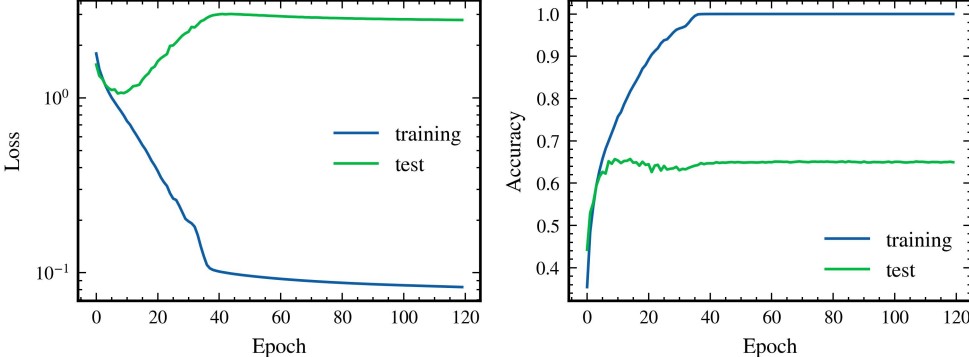

Figure 8: The evolution of the loss (left) and accuracy (right) during training of LeNet described in the main text. The statistics are shown for both training and test set. For the first 100 epochs, the exponential learning rate decay was used, and for the last 20 epochs, the learning rate was fixed at the final value of the exponential decay.

# I   TESTING DIFFERENT HYPERPARAMETERS

In this section we examine the dependence of the theoretical predictions on the three hyperparameters learning rate $\eta$, momentum $\beta$ and batch size $S$. For this, we train the LeNet again for 100 epochs, using an exponential learning rate schedule that reduces the learning rate by a factor of $0.98$ every epoch and afterwards we perform the numerical analysis as described in the main text.

However, we now train the network several times, always varying one of the hyperparameters while keeping the other two fixed. If not varied, the momentum was set to 0.90 and the batch size was set to 64. To ensure that training is always successful and 100% training accuracy is achieved, the initial learning rate was set to 0.005 when the batch size is varied and to 0.02 when the momentum is varied. Five different values are examined for each hyperparameter. To investigate the dependencies on the learning rate, the values 0.005, 0.01, 0.02, 0.03, and 0.04 were used for training. For momentum, the values 0.00, 0.50, 0.75, 0.90, and 0.95 were examined, and for batch size, the values 32, 50, 64, 100, and 128 were examined.

In addition, the training was repeated for five different seeds for each hyperparameter combination in order to obtain reliable results. This results in a relatively high computational cost. To reduce this, for the analysis in this section the weight and velocity variances are examined only in the subspace of the 2,000 largest Hessian eigenvalues and associated eigenvectors.

Figure 9 shows as an example the weight and velocity variances as well as the correlation times for different values of the batch size for one training seed each. It can be seen that the theory is not only valid for the hyperparameter combination from the previous section, but is also generally applicable for different hyperparameters. In particular, we see that there is still good agreement with the theory

even if the strictly necessary condition for the theoretical derivation of the noise autocorrelation, that the number of batches per epoch $M = N/S$ is an integer, is not met.

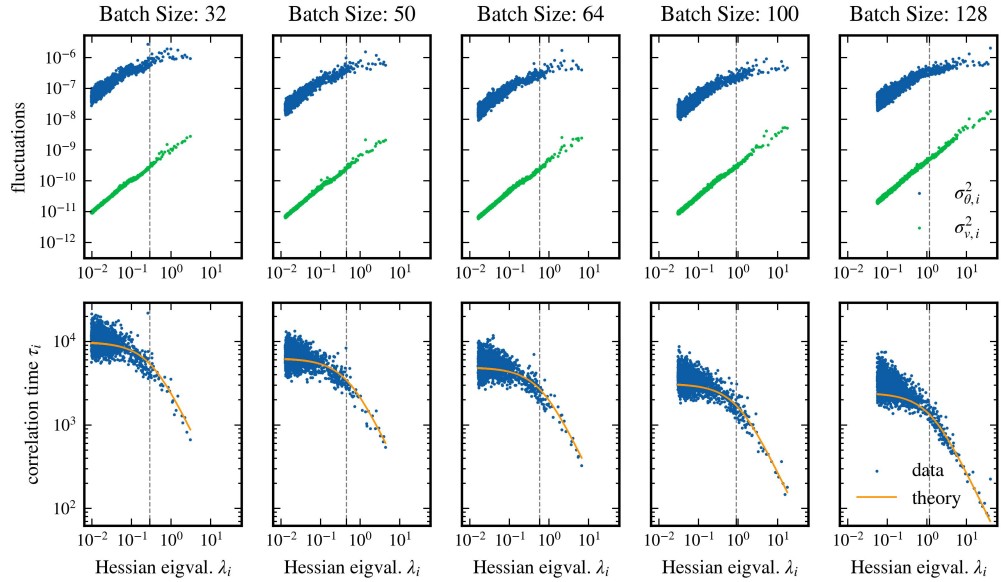

Figure 9: Testing the LeNet training with different hyperparameters. Here the relationship between the Hessian eigenvalues and the variances and correlation times for varying batch size is shown as an example. The momentum was set to 0.90 and the initial learning rate was set to 0.005.

To further examine the predictions of the theory for the hyperparametric dependencies, we now focus on the two quantities of the maximum correlation time $\tau_{\mathrm{SGD}}$ and the Hessian eigenvalue crossover value $\lambda_{\mathrm{cross}}$ and recall the theoretical predictions for these quantities:

$$\tau_{\mathrm{SGD}} = \frac{N}{3S}\frac{1+\beta}{1-\beta}\,, \tag{109a}$$

$$\lambda_{\mathrm{cross}} = \frac{3S(1-\beta)}{\eta N}\,, \tag{109b}$$

where $N$ is the number of examples in the training data set.

For the evaluation of the dependence of these variables on the hyperparameters, they were determined as follows for the various hyperparameter combinations using the data from the respective correlation time plot. For the maximum correlation time $\tau_{\mathrm{SGD}}$, the average of all correlation times was taken for which the corresponding Hessian eigenvalue is smaller than the theoretical crossover value. However, the result for the numerically determined maximum correlation time is not significantly different when simply taking the average of all determined correlation times for a hyperparameter combination, since only very few Hessian eigenvalues are larger than the crossover value.

For the numerical determination of the crossover value $\lambda_{\mathrm{cross}}$, a linear function was first fitted to the correlation times of the 20 largest Hessian eigenvalues in the log-log plot of the correlation times against the Hessian eigenvalues. In this region of the first 20 values, the correlation time is always clearly dependent on the Hessian eigenvalue and does not yet belong to the region of constant correlation times. The intersection of the fitted line with the numerically determined maximum correlation time $\tau_{\mathrm{SGD}}$ is then taken as the crossover value $\lambda_{\mathrm{cross}}$. If the numerically determined correlation times follow the theory exactly, then the correlation times determined in this way for $\tau_{\mathrm{SGD}}$ and $\lambda_{\mathrm{cross}}$ would also follow the theory accurately.

And indeed, Figure 10 shows a good agreement between the theory and the numerically determined values, although it should be noted that the deviations are larger than the random fluctuations between the different seeds.

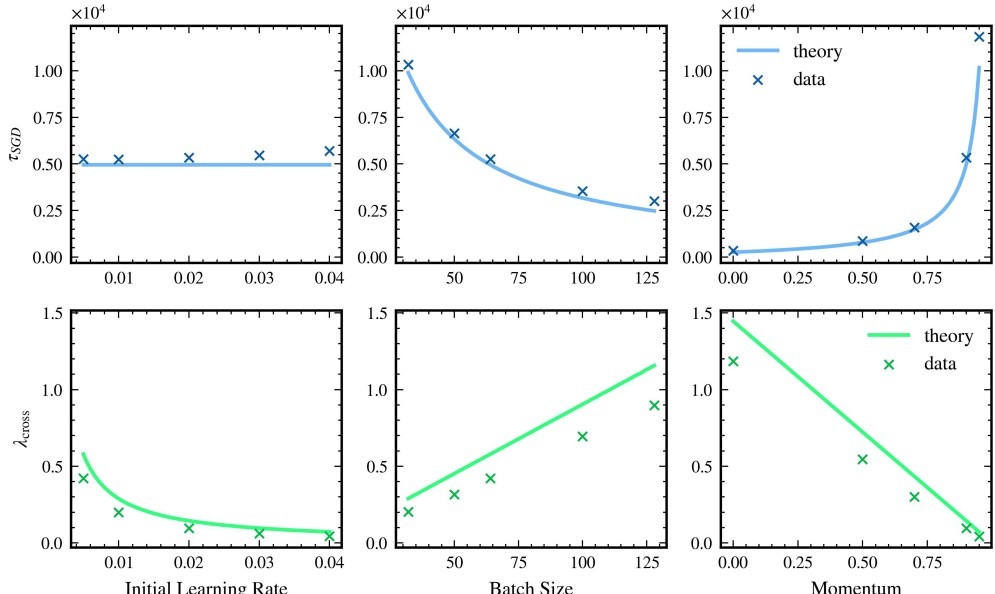

Figure 10: The empirically determined maximum correlation time, $\tau_{\text{SGD}}$, and the empirically determined crossover value $\lambda_{\text{cross}}$ for the training of the LeNet with different hyperparameters, averaged over five different seeds for each set of hyperparameters. The predictions of our theory are shown as a solid line. The fluctuations between different seeds are smaller than the marker size and are therefore not included in the figure.

## J  TEST ACCURACY AND THE HESSIAN EIGENDECOMPOSITION OF THE WEIGHT VECTOR

Recent research (Gur-Ari et al., 2018; Sabanayagam et al., 2023) suggests that the characteristics of a minimum in a neural network's loss landscape are largely determined by the large Hessian eigenvalues and their corresponding eigendirections. However, our findings indicate that flat directions, which correspond to smaller eigenvalues, should not be overlooked. When we decompose a network's weight vector in the Hessian eigenbasis, it becomes evident that both the projections onto high curvature directions and those onto relatively flat directions are important.

In Figure 11, we demonstrate this with the LeNet model. We analyzed the model's test accuracy in relation to the exclusion of projections onto Hessian eigenvectors, starting with those possessing the largest eigenvalues. After calculating the top 10,000 Hessian eigenvalues and eigenvectors out of a total of 137,000, we observed that omitting the projection onto the top 35 eigenvectors (those above the crossover value $\lambda_{\text{cross}}$) resulted in a decline in test accuracy from 63% to 54%. Discarding the top 5,000 eigenvectors, as analyzed in our main study, further reduced accuracy to 44%. It was only after excluding around 7,000 eigenvectors that the accuracy plummeted to 10%, akin to random guessing. This underscores the significance of the weight vector's orientation relative to flat Hessian eigendirections for maintaining test accuracy, highlighting the potential impact of weight fluctuations in these directions.

The pronounced accuracy drop observed after removing around 6,000 eigenvectors may be attributed to the unexpectedly large projection of the weight vector onto these eigenvectors, which is apparent from Figure 11.

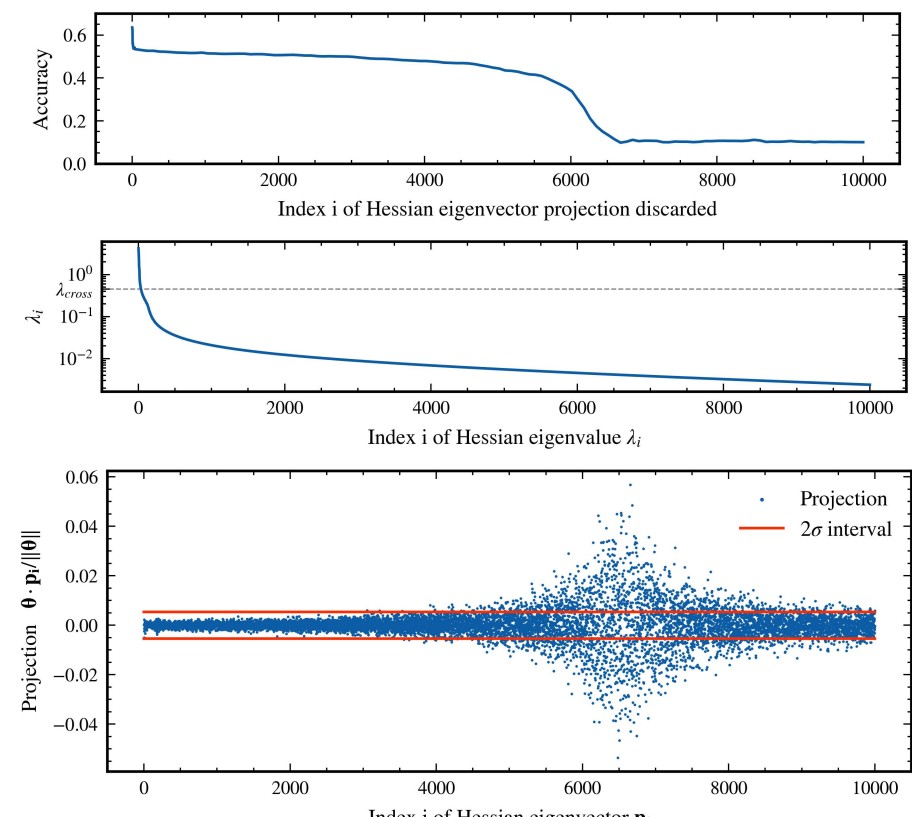

Figure 11: The top panel shows the test accuracy of the LeNet discussed in the main text and how it changes as we cumulatively discard projections onto more and more Hessian eigenvectors, starting with those with the largest eigenvalues. There is a sharp drop in accuracy for the first few eigenvectors, and then only after more than 6,000 discarded projections out of a total of 137,000 does the accuracy drop rapidly down to 10%, which is equivalent to random guessing. For comparison, the middle panel shows the corresponding Hessian eigenvalues and compares them to the crossover value $\lambda_{\text{cross}}$, where the lower than expected weight variances hold only for eigendirections with eigenvalues smaller than this crossover value. It is in the range of eigenvalues significantly smaller than the crossover value that the accuracy drops sharply to 10%. The bottom panel shows the decomposition of the weight vector $\theta$ in the Hessian eigenbasis. One explanation for the rapid drop in accuracy could be that the projection of the weights onto the corresponding eigenvectors is considerably larger than one would expect if the weight vector was a random vector. The expected $2\sigma$ interval for a random vector is displayed in red.

## K    DIFFERENT NETWORK ARCHITECTURE

To further confirm our theoretical predictions within trained networks, in this section we turn to a more modern architecture. Instead of the previously used LeNet network, we examine the ResNet-20 network (He et al., 2016). It is a convolutional network with significantly more convolutional layers than LeNet. It also uses residual blocks with residual connections, which allows for deeper network structures. As our loss function, we again employed Cross Entropy, along with an L2 regularization with a prefactor of $10^{-4}$ and we did not use batch normalization. The number of layers, which is already indicated in the name with 20, is significantly higher than in the LeNet with just five layers. With approximately 272,000 parameters, the ResNet-20 also has significantly more parameters and the computational cost is significantly higher.

Therefore, in this section we limit ourselves to examining the weight and velocity variances in the subspace of the 400 largest Hessian eigenvalues and associated eigenvectors. The network was trained with SGD for 100 epochs using the same exponential learning rate schedule as before, with

a learning rate of $5 \cdot 10^{-3}$, a momentum parameter of $0.9$, and a minibatch size of $50$. This setup achieves 100% training accuracy and 73% testing accuracy. In Figure 12 one can observe a good agreement between the theory predictions for the variances and correlation times and the numerical observations.

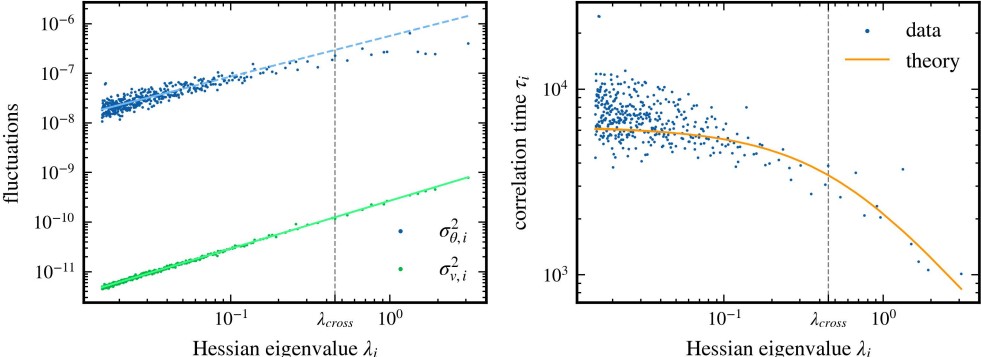

Figure 12: Relationship between Hessian eigenvalues and the variances and correlation times for a ResNet-20 trained on CIFAR10. The mean velocity of the weight trajectory was subtracted. The solid lines in the left panel indicate the regions utilized for a linear fit. The exponents resulting from the power law relationship are $0.820 \pm 0.099$ for weight variance and $0.965 \pm 0.005$ for velocity variance, with a $2\sigma$-error. The analysis was performed for the 400 largest Hessian eigenvalues and corresponding eigendirections.

## L  RELATION BETWEEN HESSIAN AND NOISE COVARIANCE

The exact variance equation of Theorem 4.3 described in Section 4.3 and the calculation shown in Appendix B are still valid even if the previously mentioned assumption $[\mathbf{C}, \mathbf{H}] \neq 0$ is not given. However, the calculated weight and velocity variances are no longer eigenvalues of the corresponding covariance matrices, but variances in the directions of the chosen eigenvector of the Hessian matrix.

Assuming that $\mathbf{C}$ and $\mathbf{H}$ do not necessarily commute, we can still project the update equations onto an arbitrary Hessian eigenvector $\mathbf{p}_i$ with eigenvalue $\lambda_i$, which gives us

$$v_{k,i} = -\eta \lambda_i \theta_{k-1,i} + \beta v_{k,i} - \eta \mathbf{p}_i \cdot \boldsymbol{\delta g}_k \,, \tag{110a}$$

$$\theta_{k,i} = (1 - \eta \lambda_i)\theta_{k-1,i} + \beta v_{k,i} - \eta \mathbf{p}_i \cdot \boldsymbol{\delta g}_k \,. \tag{110b}$$

Since we assume that $\mathbf{p}_i \cdot \boldsymbol{\delta g}_k$ is independent of weights and velocity, these two equations are decoupled for each individual Hessian eigenvector. As $\mathbf{p}_i \cdot \boldsymbol{\delta g}_k$ still follows the proposed anti-correlation, the calculations of Appendix B can be performed similarly. Therefore, in a case without commutativity, the theory makes predictions for the variances along the Hessian eigenvectors, $\sigma_{\boldsymbol{\theta},i}^2 := \mathbf{p}_i^\top \boldsymbol{\Sigma} \mathbf{p}_i$ and $\sigma_{\mathbf{v},i}^2 := \mathbf{p}_i^\top \boldsymbol{\Sigma_v} \mathbf{p}_i$, depending on the noise variance in the given direction, $\sigma_{\boldsymbol{\delta g},i}^2 := \mathbf{p}_i^\top \mathbf{C} \mathbf{p}_i$, with $\sigma_{\boldsymbol{\delta g},i}^2 = \left\langle (\mathbf{p}_i \cdot \boldsymbol{\delta g}_k)^2 \right\rangle$. The results imply that the weight covariance restricted to the Hessian eigenspace corresponding to eigenvalues smaller than the crossover value, $\lambda_i < \lambda_{\mathrm{cross}}$, denoted by $\boldsymbol{\Sigma}_<$, is reduced compared to a setup without anti-correlations, independent of whether $\mathbf{C}$ and $\mathbf{H}$ commute. For example, when considering the trace of this restricted weight covariance, $\mathrm{Tr}(\boldsymbol{\Sigma}_<)$, the theory predicts the following reduction,

$$\frac{\mathrm{Tr}(\boldsymbol{\Sigma}_{<,\text{with anti-correlations}})}{\mathrm{Tr}(\boldsymbol{\Sigma}_{<,\text{without anti-correlations}})} \approx \frac{\frac{M\eta}{3(1-\beta)} \sum_{i=i_{\mathrm{cross}}}^d \lambda_i \sigma_{\boldsymbol{\delta g},i}^2}{\sum_{i=i_{\mathrm{cross}}}^d \sigma_{\boldsymbol{\delta g},i}^2} \,, \tag{111}$$

where $i_{\mathrm{cross}}$ is the index of the largest Hessian eigenvalue $\lambda_i$ smaller than the crossover value $\lambda_{\mathrm{cross}}$.

Furthermore, the theory prediction for the correlation time $\tau_i$, defined as the ratio between the weight and the velocity variance in the eigendirection $\mathbf{p}_i$ of the Hessian, is also valid independently of the

commutation relation between $\mathbf{C}$ and $\mathbf{H}$. However, the weight and the velocity variance in the eigendirection $\mathbf{p}_i$ defined above are no longer eigenvalues of the covariance matrices $\Sigma$ and $\Sigma_\mathbf{v}$ if $\mathbf{p}_i$ is not also an eigenvector of $\mathbf{C}$. Nevertheless, these variances are still a good approximation for the actual eigenvalues, as long as $\mathbf{C}$ and $\mathbf{H}$ commute approximately.

While empirical investigations show that $\mathbf{C}$ and $\mathbf{H}$ do not commute exactly, they show a high alignment and suggest that the Hessian eigenbasis is a reasonable approximation for the eigenbasis of the noise covariance (see Section 3.1). We find such an approximate commutativity as well. We investigated the Hessian and the covariance of the recorded noise in the approximated eigenspace of the Hessian during the analysis period of the LeNet considered in the main text. In Figure 13 one can see, that the actual 200 largest eigenvalues of the noise covariance align well with the variances in the directions of the 200 Hessian eigenvectors corresponding to the largest Hessian eigenvalues, indicating that the Hessian eigenbasis is very similar to the eigenbasis of the noise covariance.

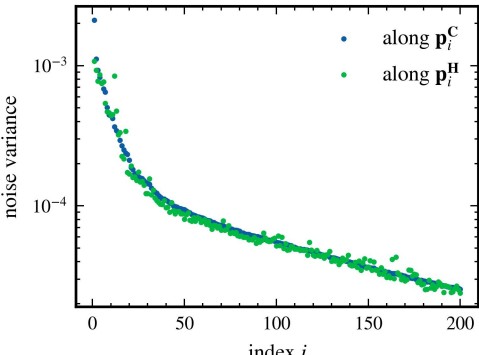

Figure 13: Noise variance during the analysis period of the LeNet considered in the main text. We analyzed the gradient noise recorded in the subspace of the 5,000 largest Hessian eigenvalues and corresponding eigenvectors $\mathbf{p}_i^\mathbf{H}$. We calculated the covariance matrix of the recorded noise, $\mathbf{C}$, and its eigenvectors $\mathbf{p}_i^\mathbf{C}$ in descending order according to their eigenvalues. We then plotted the largest eigenvalues of the noise covariance, which is the noise variance along $\mathbf{p}_i^\mathbf{C}$, together with the noise variance along $\mathbf{p}_i^\mathbf{H}$. When $\mathbf{C}$ and $\mathbf{H}$ commute, there exists a simultaneous eigenbasis for both matrices, and the shown variances would align perfectly. We find that the variances show very good alignment, indicating that both matrices commute approximately.

We also investigated how well the approximation of proportionality between $\mathbf{C}$ and $\mathbf{H}$ holds. For this we calculated the cosine similarity between both matrices in the approximated Hessian eigenspace, which is the normalized dot product between the flattened matrices. We found a cosine similarity of 0.82, which is similar to the empirical similarities found by Thomas et al. (2020) and significantly higher than that of two random low-rank matrices. Additionally, we plotted the noise variances along the approximated Hessian eigenvectors against the corresponding Hessian eigenvalues and found that the two indeed follow an approximate linear relationship (see Figure 14). So, while an exact proportionality between $\mathbf{C}$ and $\mathbf{H}$ is not satisfied, it seems to be a good approximation.

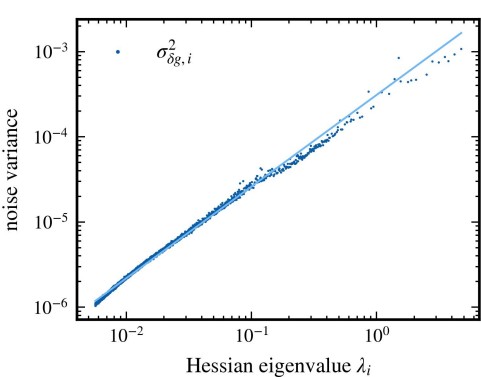

Figure 14: Relationship between Hessian eigenvalues and the variances of the recorded noise during the analysis period of the LeNet considered in the main text. The solid line signifies a linear fit. The exponent resulting from the power law relationship is $1.075 \pm 0.002$ with a $2\sigma$-error. If the noise covariance were proportional to the Hessian matrix, the exponent should be equal to one.

