# OpenReview forum: "Anti-Correlated Noise in Epoch-Based Stochastic Gradient Descent: Implications for Weight Variances"
_ICLR.cc/2025/Conference — Submitted to ICLR 2025_

### Official Review · Reviewer_2rHA · 2024-10-25

**Soundness:** 3
**Presentation:** 4
**Contribution:** 2
**Rating:** 5
**Confidence:** 3

**Summary:**

This paper explores the consequences of anti-correlations of update steps which arise due to selection without replacement in SGD. The paper rests on the assumption that we are at the end of training, or that we are near enough a minimum of the loss that the landscape is quadratic. The analysis begins by characterizing the different-time covariance of the gradients in terms of the covariance at equal times, and verifying numerically that their formula is correct. They use this to derive the related formulas for the covariance of the weights, and the covariance of the updates to the weights.

They define a correlation time, $\tau_i$ as the ratio of these two covariances, which becomes the central object of study. Using the results from before, they calculate $\tau_i$ in terms of the momentum hyperparameter, $\beta$ and hessian eigenvalue, $\lambda$, which reveals two distinct limits. When the eigenvalue is large they find that the correlation time decreases as $\lambda^{-1}$ while it is constant when $\lambda$ is small. These results are all confirmed numerically for the top 5000 eigenvalues of the spectrum of a LeNet model on CIFAR10.

**Strengths:**

This paper is clearly written with a transparent analysis. Mathematical analysis is generally paired with sufficient words to explain the meaning of the relevant equations, and is well-motivated in itself. They perform a deep analysis in a relatively common setting of sampling without replacement, which should carry over (at least heuristically) to all such algorithms. Though the experiments are at a small-scale this is not an issue in my opinion as the result is mathematically robust, and the experiments are illustrative. Because the setting is well-circumscribed, the applicability of this work could be tested on a case-by-case basis.

**Weaknesses:**

The primary weakness, which is why I did not recommend this paper for acceptance, is my concern about the relevance of this work to the practical setting. In my reading of the paper I did not understand the motivation behind studying the end point of training as a limit. Due to this, the contribution of this paper is limited due to the specific setting considered. I am glad to increase my rating if the authors could sufficiently clarify the following two questions:

1. How can we understand finite training time behavior from this analysis?
2. In the setting where some $\lambda_i = 0$ exactly, how can we make sense of the constant weight assumption?

**Questions:**

These questions are lower-priority. Answers to them are primarily towards understanding the implicit viewpoint of the paper

1. What happens in the case of a loss function like $L(x) = x^4$ which is not well-described by a quadratic approximation near its minimum?
2. Orvieto et al. 2022 consider adding noise which is independent from the data rather than data-dependent SGD noise. How can we know that the anti-correlations in that kind of noise behave the same way as anti-correlations in SGD noise?
3. Is it true that sampling without replacement is better than increasing batch size, assuming that both result in the same overall gradient variance?
4. How do you justify subtracting the weight drift at late times for experiments when your assumptions don't require this?

---

> ### Author Response · Authors · 2024-11-22
> **Response to Reviewer 2rHA (1/2)**
>
> We thank the reviewer for their insightful feedback and are pleased to address your comments and questions in detail below.
>
> ---
> *Regarding finite training time behavior:*
>
> Our analysis, while focusing on behavior near a minimum, provides insights that are relevant even during finite training times. When a network has been trained for only a few epochs, its parameters may be close to a minimum in some directions (those with large Hessian eigenvalues), but there remain many relatively flat directions where the loss can decrease further. In these flat directions, the lower-than-expected weight variance in the Hessian eigendirections with small eigenvalues allows gradients to dominate over noise-induced fluctuations. This can potentially steer the network toward even flatter minima.
>
> To illustrate this, consider the "widening valley" scenario as a prototypical loss function:
>
> $L(\mathbf{u}, v) = \frac{1}{2} v^2 \|\mathbf{u}\|^2, \quad v \in \mathbb{R}, \quad \mathbf{u} \in \mathbb{R}^N.$
>
> This loss is flat in all directions except one, where the curvature $\lambda$ depends on the position in other directions. Orvieto et al. (2022) proved that in this setup, gradient descent with anti-correlated noise moves toward flatter regions, while uncorrelated noise increases curvature.
>
> Anti-correlations in the noise occur if the noise terms  $\delta g_k(\theta) = g_k(\theta) - \nabla L(\theta)$ change minimally over one epoch, which is reasonable even during finite training since weights often change slowly over an epoch. Crucially, it's not necessary for the weights to remain completely constant; minimal changes suffice for the anti-correlated effects to manifest. Therefore, our analysis is applicable and provides meaningful insights into finite training time behavior.
>
> ---
> *Relatively constant weights and flat directions:*
>
> When some Hessian eigenvalues are zero, corresponding to perfectly flat directions, the gradient offers no force to move the parameters along these directions. Under our assumption of relatively constant weights over an epoch, the anti-correlations in the noise become particularly significant in these flat directions.
>
> Even with $\lambda_i = 0$, anti-correlated noise prevents the parameters from diffusing freely, which would be the case with uncorrelated noise. Instead, the anti-correlations ensure that the variance of the parameters remains bounded. Specifically, if the noise covariance is $\sigma^2$ and the correlation time is $\tau$ (on the order of a few update steps), the parameter variance becomes $\langle x_t^2 \rangle \propto \tau \sigma^2$, independent of the number of update steps after an initial period. This contrasts with the unbounded growth ($\propto t$) that occurs with uncorrelated noise.
>
> Therefore, even when $\lambda_i = 0$, assuming relatively constant weights over an epoch allows us to understand how anti-correlated noise influences the training dynamics, keeping the parameter variance finite and the analysis meaningful.
>
> ---
> *Question 1:*
>
> Our analysis is based on the assumption of a quadratic loss near the minimum. For loss functions like $L(\theta) = \theta^4 $, which are not well-approximated by a quadratic near their minimum, the specific results of our analysis may not directly apply. However, we expect the parameter variance to depend on the noise variance $\sigma^2$ in a more complex manner: 1. For sufficiently small $\sigma^2$, our original results for flat directions, where the variance is limited by anti-correlations rather than curvature, should hold approximately. 2. For larger $\sigma^2$, where the predicted parameter variance for flat directions becomes comparable to or exceeds the width of the quartic minimum (order 1 or larger in this case), the variance would instead be constrained by the loss shape. In this regime, we suspect that the parameter variance would resemble that of a quartic loss without noise anti-correlations. We believe this distinction could provide a basis for future analysis of non-quadratic losses.
>
> ---
> *Question 2:*
>
> While the noise in Orvieto et al. (2022) differs in covariance structure from SGD noise, the temporal correlations are analogous to those in our analysis (specifically in Theorem 4.1). The anti-correlations they introduce have a short correlation time, similar to the noise correlations in SGD without replacement when the number of batches is small. The key similarity is that both types of noise exhibit anti-correlations over time, which can influence the training dynamics in comparable ways. This suggests that insights from their analysis are relevant to understanding the effects of anti-correlated SGD noise.

---

> ### Author Response · Authors · 2024-11-22
> **Response to Reviewer 2rHA (2/2)**
>
> *Question 3:*
>
> Our study did not directly compare the effects of sampling without replacement to increasing the batch size regarding their impact on loss or generalization. As such, we cannot make definitive statements about which approach is better under the assumption of equal overall gradient variance. This is an interesting question that merits further investigation, and we acknowledge it as a potential avenue for future work.
>
> ---
> *Question 4:*
>
> Our theoretical assumptions focus on the parameters being very close to the minimum of a quadratic loss, implying negligible drift. However, in practice, even after extensive training, parameters may still be far from an ideal quadratic minimum due to the complex nature of the loss landscape, which might resemble scenarios like the "widening valley" described earlier. These complexities can induce weight drifts not accounted for in a simple quadratic approximation. In our experiments, we subtracted the weight drift at late times to isolate the effects predicted by our theory from these additional sources of drift.

---

> > ### Comment · Reviewer_2rHA · 2024-11-26
> > **Response to Authors**
> >
> > I thank the authors for their answers to my questions. I have some follow up questions:
> >
> > **Question 2:**
> > As I understand it, you suggest that your analysis and the analysis of Orvieto et al. are similar enough because of the similarity in temporal structure, and that the structure of the noise itself doesn't matter as much. On the other hand SGD is known to produce different stationary distributions, in particular those with heavy tails [1] compared with isotropic Gaussian noise which sounds strongly different. My intuition is that this would tend to enhance the difference between the two settings you consider, so maybe it is okay. Do the authors agree?
> >
> > **Question 4:**
> > So would you agree that even in your experiments you're not in a regime where your theory is mathematically valid? Is you claim that the theory extrapolates to the settings considered in experiment? This makes me a bit skeptical, because even in your carefully designed experiments you are not able to reach the setting analyzed mathematically.
> >
> >
> > **References:**
> >
> >
> > 1. Gurbuzbalaban, Mert, Umut Simsekli, and Lingjiong Zhu. "The heavy-tail phenomenon in SGD." International Conference on Machine Learning. PMLR, 2021.

---

> > > ### Author Response · Authors · 2024-11-28
> > >
> > > Thank the reviewer for their follow-up question and we are happy to further clarify our perspective.
> > >
> > > **Question 2:** We appreciate the reviewer’s observation. Without considering anti-correlations, the noise covariance typically observed in SGD causes the stationary weight covariance to be isotropic. In contrast, for isotropic noise covariance, the weight covariance in flat directions would be larger than in steep directions. In this sense, we agree that SGD noise amplifies the effect of reducing the weight variance in flat directions when compared with steep directions, similar to how anti-correlations reduce weight variance in flat directions. Thus, while the temporal structure is the key similarity in our analysis and that of Orvieto et al., the specific noise characteristics of SGD might further reinforce the observed effects.
> > >
> > >
> > > **Question 4:** We find empirically that the continued evolution of weights occurs in a low-dimensional subspace, specifically, in our case, a one-dimensional subspace. This drift can be accounted for by subtracting a linear regression, after which we find clear evidence for stationary fluctuations of the weights. While we acknowledge that there are mild deviations from the strict mathematical assumptions of our theory, we see it as a strength that the theory still accurately describes the behavior of weight variance in our experiments, showing very good agreement in a realistic scenario despite these deviations.

---

### Official Review · Reviewer_pnyq · 2024-10-28

**Soundness:** 3
**Presentation:** 3
**Contribution:** 3
**Rating:** 6
**Confidence:** 4

**Summary:**

The authors study the dynamics of SGD in a discrete-time regime while sampling without replacement which leads to correlations in the noise between different time steps. By invoking certain assumptions, they derive the correlation function of the noise and claim that the anti-correlations in the noise may cause better generalization ability.

**Strengths:**

This paper is well-organized and well-written. The definitions and assumptions are made clear. The auto-correlation function of the gradient noise between different time steps is an interesting topic. The theoretical findings are solid upon the assumptions and experiments are thorough.

**Weaknesses:**

See **Questions**.

**Questions:**

While the overall quality of this paper is good, there are still several questions I would like to ask.

1. The main theorem is derived on the basis of three fundamental assumptions 1-3. Assumption 1 is a commonly adopted one if the discussion is limited to a local minimum. However, **Assumption 2** is newly invoked by the authors on the state-dependence of the noise. The validity of this assumption is discussed through mere words and a "look" at the experiment done in Appendix H.  It lacks a rigorous demonstration of when this assumption can be adopted and how large deviations it may lead to the main theorems. Also, the experiment and discussion provided in Appendix H are extremely specific to the very task, and a detailed connection to Assumption 2 is appreciated.

2. **Assumption 3** is about the commutation relation between the noise covariance and the Hession. The authors claim that "this assumption is not strictly necessary but it simplifies the analysis". They provide evidence in Appendix L that under the circumstances they are considering, these 2 matrices are almost aligned with each other. In Appendix L they calculate a 0.82 cosine similarity between them and claim that "it seems to be a good approximation" in the final sentence. The details in this calculation are not provided and a 0.82 similarity does not seem satisfactory to claim that two things are alike in a common sense.  Besides, "it seems to be a good approximation" sounds like the authors themselves are not very confident in this statement and it is not a rigorous manner to make an approximation. I would like the authors to elaborate more on this assumption. Could the authors provide more details about the cosine similarity of two matrices? What level of similarity would be considered sufficient? Is it possible to provide a quantitative analysis of the influence of this cosine similarity on the validity of Approximation 3? Finally, a minor question, if this assumption is "not strictly necessary", is it possible to just discard it and establish a more general theory?

3. The experiments are performed on CIFAR10 using the LeNet and the details are provided in Appendix H. From **Figure 8**, it seems that the model is severely overfitting and the authors make no comments about overfitting. Is overfitting related to any assumptions made in the paper? I'm curious about that is this severely overfitting situation chosen intentionally. If yes, why? If not, what will happen to the results if the model does not overfit?

---

> ### Author Response · Authors · 2024-11-22
> **Response to Reviewer pnyq**
>
> We thank the reviewer for their valuable feedback, and we provide detailed responses to their questions below.
>
> ---
> *Question 1:*
>
> We want to note that it should be sufficient to encounter approximate state-independence only within each epoch individually to empirically observe variance reduction. Furthermore, it is common in many training tasks for the loss to change only minimally over the course of an epoch, enabling such approximate state-independence.
>
> ---
> *Question 2:*
>
> First, we would like to note that the assumption of high alignment is commonly made in the literature and has been rigorously studied (e.g., see [Thomas, 2020]), so we refer the reader to these works for a more detailed investigation of this topic.
>
> Regarding our specific case, consider two symmetric $N \times N$  matrices of rank one, of the form $uu^\top$ (with entries of $u$ being standard normally distributed). Their cosine similarity is on average $1/N$. For low-rank matrices constructed as sums of $M$ such random matrices, the cosine similarity is on average $M/N$. In our case, we considered matrices with $N=5000$ and only about 10-20 outliers in addition to the small bulk of values. Without any connection between the matrices, one would expect a cosine similarity of approximately 0.004. Therefore, a cosine similarity of 0.82 indicates a very high alignment.
>
> Finally, we also want to clarify that the assumption of commutation between $H$ and $C$ is not strictly necessary for our analysis. As shown in Appendix L, our framework still predicts variance reduction in the Hessian eigenbasis even without this assumption, and we can derive results, such as a reduced trace of the weight covariance, independently of the commutation relation. However, we chose to present our results under this assumption because it is a commonly made assumption, it aligns well with the empirical situation (as demonstrated, among other arguments, by the high cosine similarity we calculated), and it simplifies the interpretation of our results.
>
>
> [Thomas.2020] - arXiv:1906.07774
>
> ---
> *Question 3:*
>
> It is common to observe a higher training accuracy than test accuracy for the CIFAR10 dataset, and the training schedule was inspired by previous studies. Additionally, we obtained similar results with a ResNet architecture, as described in the appendix, which exhibited a smaller gap between training and test accuracy. We do not believe that the results are specific to the training schedule, and we expect them to be replicable for any reasonable set of parameters close to a potential minimum of the loss.

---

> > ### Comment · Reviewer_pnyq · 2024-11-26
> >
> > I thank the authors for their reply. However, as I stated in the review, assumptions should be made with more rigor rather than by wording. The reply does not address my concerns thoroughly and I cannot raise my scores. I decide to keep my score as it was.

---

### Official Review · Reviewer_oKMV · 2024-11-03

**Soundness:** 2
**Presentation:** 2
**Contribution:** 3
**Rating:** 6
**Confidence:** 3

**Summary:**

The paper analyzes the behavior of SGD(m) without replacement (RandomReshuffle) and its implicit bias when the iterates reach a basin of some minimum. Theoretically, they derive the exact anti-correlation of the gradient noise for static weight. Then, under additional assumptions, they prove a connection between such anticorrelations and the variance of the weight fluctuations. Qualitatively speaking, this implies that the weight variance is decreased *only* in the flat directions, showing that without replacement provides a benign implicit bias towards flat minima. Their theoretical results are complemented with various experiments.

**Strengths:**

- An important gap in the SGD dynamics literature was considered by considering batched SGD without replacement in finite learning rate, showing explicit geometry-dependent implicit bias compared to replacement.
- The theory is general enough to include momentum SGD.
- Sufficient experimental results supporting the theories

**Weaknesses:**

- IMHO, the biggest point preventing me from giving a higher score (along with my questions below) is that *some (not all)* of this paper's main conclusions seem to overlap with the preprint [1] for SGD without replacement (without momentum). Given that the authors could provide a satisfactory answer, I'm open to raising my score further.
To my knowledge, [1] showed that in-expectation, for both **ShuffleOnce** and **RandomReshuffle**,
    - SGD without replacement = SGD with replacement along larger curvature "+" Shrinking the tr(cov of gradients) along flatter directions (see their Theorem 1; "+" means decoupling)
    - Actually, SGD without replacement adds an implicit regularize that penalizes a weighted tr(cov of gradients) over single data points (see their Theorem 3)
    - In all their derivations, they do not make any restrictive assumptions (e.g., quadratic loss, anti-correlated noise for changing state, $[C, H] = 0$, etc)

- (continuing from above) All in all, given how the problem that [1] is tackling (behavior of SGD without replacement) and the qualitative general conclusion (greater curvature is similar to uncorrelated noise (with replacement), and flatter curvature decreases the fluctuation) both seem similar, I'm curious on what additional intuitions and results this paper adds compared to [1], and how the results presented here are related to [1]. I feel this should be much more emphasized, given that this paper relies on several assumptions that [1] does not consider: quadratic loss, anti-correlated noise even under dynamically changing weights, and $[C, H] = 0$. Especially the last assumption is critical for all of the theories presented in this paper, which allows for the use of common eigenbasis.

- (continuing from above) Immediately I see two advantages: one is that this paper's framework encompasses momentum and that this has more realistic experiments supporting the theories. I'm looking for something like "under such additional assumptions, our paper provides a much more precise characterization of ..... compared to [1], which can only say that ....".

- [writing] Overall, the writing and organization should be improved. Many important discussions have been completely relegated to the Appendix. Space-wise, maybe move/reduce Sec 3?

- [writing] The authors should consider making all the assumptions explicit in the main text and collecting them in an orderly fashion, instead of using phrases such as "under general assumptions, stated in Appendix C" or ""With the above assumptions".


[1] https://arxiv.org/abs/2312.16143

**Questions:**

- I'm a bit confused about the statement, "Next, we consider the probability that two batches $k$ and $k + h$, separated by $h$ update steps, belong to the same epoch". I understood this as follows: given two update step indexes $k$ and $k + h$, what is the probability of the corresponding batches belonging to the same epoch? I don't see any randomness in this statement. Yes, the batches are random, but if the indices are deterministically given, then the two indices are in the same epoch if there exists a $i \geq 1$ such that $M (i - 1) < k < k + h \leq M i$. I guess the randomness is supposed to be w.r.t. the randomness of the batches being sampled, but
  - In my head, $\frac{M - |h|}{M}$ is the probability of two deterministic batches being in the same epoch when we uniformly randomly allocate their indices such that they are separated by $h$ steps.
  - Maybe I'm completely missing (or misunderstanding) something here, so please feel free to correct me here!
  - As a clarification, you are considering RandomReshuffling, where at each epoch, an ordered partition of $[N]$ is sampled at random, right? The author should consider making this precise, as there is another variant of without replacement sampling, namely, ShuffleOnce, where a single shuffle takes place at the beginning and the same batches are used in the same order for all the epochs.

- Although I agree that weights remain approximately constant near minima, theoretically, could one use appropriate perturbation arguments (e.g., Taylor expansion) to provide a more complete theory of anti-correlation with varying states? Or would such more intricate analyses not give any useful insights and thus be unnecessary?

- Does the current analysis extend to ShuffleOnce, or at least empirically?

- [low priority, did not affect my initial evaluation] If time allows, can authors try the similar experiments for small scale transformer?

- Section 4.2 is IMHO too sudden without proper motivation. Why are we suddenly interested in the weight and velocity variance? Why consider their ratio? What is the intuition of the quantity *correlation time*? Should it be understood as, 'after the correlation time, the weights and velocities are somehow correlated'...?
Moreover, Section 4.2 refers to the setup described in Section 4.3, making me think that for the sake of organization, Section 4.2 should come after 4.3?

---

> ### Author Response · Authors · 2024-11-22
> **Response to Reviewer oKMV (1/2)**
>
> We thank the reviewer for their valuable feedback, and we provide detailed responses to their comments and questions below.
>
> ---
> *Relationship to Beneventano (2023) [arXiv:2312.16143]:*
>
> We thank the reviewer for bringing [1] to our attention and for the opportunity to clarify how our work relates to this preprint. While we conducted our research independently of [1], we recognize the importance of situating our contributions within the context of related work.
>
> Reference [1] investigates the behavior of SGD without replacement, showing that, in expectation, both ShuffleOnce and RandomReshuffle schemes cause SGD without replacement to behave like SGD with replacement along directions of larger curvature, plus an effect that reduces the trace of the covariance of gradients along flatter directions (as stated in their Theorem 1). They also demonstrate that SGD without replacement introduces an implicit regularizer that penalizes a weighted trace of the covariance of gradients over individual data points (Theorem 3). Notably, their analysis does not rely on restrictive assumptions such as quadratic loss functions or anti-correlated noise.
>
> Our work complements and extends these findings in several key aspects. We incorporate momentum into our framework, analyzing how it interacts with the anti-correlation introduced by sampling without replacement. This allows us to provide a more precise characterization of the optimization dynamics under momentum, which is prevalent in practical deep learning applications and directly applicable to real-world scenarios.
>
> While [1] focuses primarily on the covariance of gradients, we extend the analysis to the covariance of the weights themselves. By providing precise results on how sampling without replacement affects the weight covariance, including variance reduction effects, we offer a deeper understanding of the weight dynamics that contributes to insights into convergence behavior and generalization performance.
>
> Although our analysis involves certain assumptions -- such as a quadratic loss approximation and considerations of anti-correlated noise -- we show that some of our key results do not strictly depend on these assumptions. Specifically, we demonstrate in Appendix C that variance reduction in the Hessian eigenbasis can be predicted without requiring the commutation of the Hessian matrix $H$ and the covariance matrix $C$, broadening the applicability of our framework beyond the initial assumptions.
>
> We also note that an earlier version of our work was made publicly available on arXiv six months prior to the appearance of [1], and unfortunately, it was not cited there. This suggests that both works were developed independently, and our findings precede those presented in [1].
>
> Furthermore, our paper includes extensive experiments that support our theoretical findings, demonstrating that our assumptions and derived results hold in practice across various neural network architectures and datasets. This empirical backing strengthens the practical relevance of our contributions and provides evidence that our more precise characterizations offer tangible benefits over the broader conclusions in [1].
>
> In summary, while [1] provides valuable insights into the behavior of SGD without replacement, our work builds upon and extends these ideas by encompassing momentum, analyzing weight covariance in detail, and relaxing certain assumptions to increase the generality of our findings. By validating our theories through realistic experiments, we enhance the credibility and applicability of our work.
>
> [1] - arXiv:2312.16143.
>
> ---
> *Question 1:*
>
> For the probability that two batches $k$ and $k + h$, separated by $h$ update steps, belong to the same epoch, we consider the average over $k$, since in the end we also consider a covariance averaged over update steps $k$. To attain the same probability we anticipate, one could also ask, given any batch $k$ within a given epoch with equal probability, what is the probability that it is one of the last $h$ batches in that epoch.
>
> ---
> *Question 2:*
>
> Given the current setup, further insights from perturbative analysis appear limited. However, we believe this question holds promise for potential future research directions.
>
> ---
> *Question 3:*
>
> The main argument described for the anti-correlations, which also facilitates the smaller than expected weight variance, is the fact that the noise terms $\delta g_k(\theta) = g_k(\theta) - \nabla L(\theta)$ over one epoch of RandomReshuffle SGD add up to zero. The same is true for ShuffleOnce SGD, therefore, we expect the same results for the weight variances.
>
> ---
> *Question 4:*
>
> We thank the reviewer for the suggestion. As our analysis is architecture independent we expect to find similar results for a transformer architecture. However, due to time restrictions, we will not be able to include such an empirical analysis.

---

> ### Author Response · Authors · 2024-11-22
> **Response to Reviewer oKMV (2/2)**
>
> *Question 5:*
>
> The correlation time described in Section 4.2 describes the velocity correlation time and indicates that there are no longer correlations in the velocity after more update steps than this timescale happened. In a quadratic loss the velocity is naturally anti-correlated as any deviation of the parameters from the minimum induced by a velocity at an initial update step is reverted by the influence of the loss, pushing the parameters back to the minimum. The timescale of this process is proportional to the curvature of the loss. The anti-correlations of the noise, however, also cause any deviation caused by the noise to be reverted after the timescale of the noise anti-correlations. The smaller of both of these time scales determines the overall behavior. We believe it is important to introduce the concept in section 4.2 as we analyze the correlation time in Section 4.3.

---

### Official Review · Reviewer_EdGc · 2024-11-03

**Soundness:** 2
**Presentation:** 3
**Contribution:** 1
**Rating:** 3
**Confidence:** 4

**Summary:**

This paper studies the correlation of gradient noise in the later training phase of SGD. They show that SGD noise will be anti-correlated over time, assuming the weights do not change. They also show that weight variance is small in flat directions, i.e., eigendirection corresponding to small eigenvalues of loss hessian. This result is obtained by assuming quadratic loss. They further implement experiments to verify their theoretical results.

**Strengths:**

1. Understanding SGD noise is an important topic in the optimization. This paper analyzes the SGD noise by connecting the covariance matrix to the Hessian of the loss and further connects the flat directions to the weight variance. I believe these insights are nice.

2. The paper is well-written and the logic is easy to follow. The experiments are nicely done to verify the theory.

**Weaknesses:**

1. I am not sure if it is reasonable to assume the fixed weights to analyze the gradient noise. It is almost trivial to obtain Theorem 4.1 under such an assumption since some probability arguments are sufficient. Additionally, if the weights are fixed, or almost fixed at the end of training, the effect of gradient noise is minimal. It should make more sense to understand the gradient noise at the beginning of training, where the weights change significantly.

2. About the assumptions in Section 4.3, I do not see why Hessian and noise covariance commute in general. It seems like they commute when the loss is quadratic in weights, which is assumed in Assumption 1. Then analyzing quadratic loss seems less interesting and I believe there should have been many results regarding this setting. Could you comment on this?

3. I suggest that the authors avoid using $\gg$ or $\approx$ in mathematical statements. These symbols are not precise, and it is unclear which terms are considered small. Furthermore, simply stating the expressions (Eqs. (8) and (9)) makes it difficult to understand how the variance of weights and velocity changes. Adding some discussion would be beneficial.

**Questions:**

1. In line 319, it is claimed that if these conditions are not met, the weight fluctuations would diverge. How to see that?

2. Is the momentum necessary in your arguments? Do the results still hold for vanilla SGD?

---

> ### Author Response · Authors · 2024-11-22
> **Response to Reviewer EdGc**
>
> We thank the reviewer for their valuable feedback, and we provide detailed responses to their comments and questions below.
>
> ---
> *Conditions for anti-correlations:*
>
> Anti-correlations in the noise occur if the noise terms  $\delta g_k(\theta) = g_k(\theta) - \nabla L(\theta)$ change minimally over one epoch, which is reasonable even during finite training since weights often change slowly over an epoch. Crucially, it's not necessary for the weights to remain completely constant; minimal changes suffice for the anti-correlated effects to manifest. Therefore, our analysis is applicable and provides meaningful insights into finite training time behavior.
>
> ---
> *Commutation of Hessian and noise covariance:*
>
> The approximation that both matrices commute is independent of the quadratic approximation of the loss and appears frequently in the literature [Jastrzebski.2017] [Zhang.2019]. The assumption is inspired by the strong alignment between the Hessian matrix $H$ and the gradient sample covariance matrix $C_0$  empirically observed for a variety of networks [Thomas.2020]. Furthermore, numerous theoretical arguments have been put forward to explain this approximate alignment [Martens.2014] [Jastrzebski.2017]. We refer to the literature mentioned in the Related Work section. The arguments mostly boil down to a decomposition of the Hessian,
>
> $H \approx M + \frac{1}{N}\sum_{n=1}^N \nabla l(x_n)\nabla^\top l(x_n)$,
>
> where the matrix $M$ can be neglected and for the gradient sample covariance matrix we have
>
> $C_0 \approx \frac{1}{N}\sum_{n=1}^N \nabla l(x_n)\nabla^\top l(x_n)$
>
> as the gradient of the total loss $\nabla L(\theta)$ is approximately zero near a minimum. Since the final noise covariance matrix $C$ is provably proportional to $C_0$, see Appendix D, we have $H \approx C \times \textrm{const.}$, which inspires the assumption $[C,H] = 0$.
>
> [Martens.2014] - arXiv:1412.1193
>
> [Jastrzebski.2017] - arXiv:1711.04623
>
> [Zhang.2019] - arXiv:1907.04164
>
> [Thomas.2020] - arXiv:1906.07774
>
> ---
> *Question 1:*
>
> If the conditions in line 319 are not satisfied, then the matrix governing the deterministic part of the update $\bf X$ (see Appendix B, Equation 18, line 761) would have eigenvalues greater than one, leading to divergence when applying multiple update steps, or equivalently, to divergence when multiplying the matrix by itself multiple times.
>
> ---
> *Question 2:*
>
> Our analysis is also valid for the momentum parameter $\beta$ being set to zero, which recovers vanilla SGD. Therefore, momentum is not necessary and the results hold for vanilla SGD as well.

---

> > ### Comment · Reviewer_EdGc · 2024-11-26
> >
> > I thank the authors for the response.
> >
> > I still do not see the point of analyzing the anti-correlation of the gradient at the end of training. If the weights almost do not change when anti-correlation in noise happens, then it suggests anti-correlation has little to do with the optimization or generalization dynamics of neural networks.
> >
> > Similarly, if the commutation of Hessian and noise covariance happens when the loss is exactly zero, I do not see why you can assume that. Is it possible to do the analysis when $[C,H] <\epsilon$ holds?
> >
> > In summary, I feel that the assumptions made in this paper are too strong and may not reflect what occurs in practice. I am inclined to maintain my score.

---

> > > ### Author Response · Authors · 2024-11-28
> > >
> > > We thank the reviewer for their response.
> > >
> > > We believe that a minimal change in weights over one epoch does not necessarily mean that there is no significant change in weights over multiple epochs. This change over multiple epochs would still allow for anti-correlations and also allow them to have an effect on the dynamics of the optimization.
> > >
> > > Furthermore, we also want to clarify that the assumption of commutation between $H$ and $C$ is not strictly necessary for our analysis. As shown in Appendix L, our framework still predicts variance reduction in the Hessian eigenbasis even without this assumption, and we can derive results, such as a reduced trace of the weight covariance, independently of the commutation relation. However, we chose to present our results under this assumption because it is a commonly made assumption, aligns well with the empirical situation, and simplifies the interpretation of our results.

---

### Meta-Review · Area_Chair_HcgH · 2024-12-19

**Metareview:**

This paper studies the anti-correlation of gradient noise for iterates of epoch-based (without-replacement) SGD and its implications to the variance of weight fluctuation at the end of training. Under the assumption that the noise of SGD is static (which holds e.g., when the noise is evaluated at the same parameter value for the entire epoch), the paper shows that the gradient noise of different iterates within a single epoch is inherently anti-correlated, and then moves on to use this result to consider the fluctuation of weights and velocity. The main message is that anti-correlation of noise reduces the fluctuation of weights along flat directions compared to with-replacement SGD. The authors also claim through experiments that this reduced fluctuation leads to better generalization performance of models trained via without-replacement SGD than with-replacement SGD.

All reviewers agreed that the paper is well-written, the considered problem is of interest, and the presented experiments align well with theory. However, at the same time, all reviewers expressed concerns about the strong assumption that the noise is static, which is essentially the same as assuming constant weight value over one entire epoch. The reviewers and I found the authors’ justification somewhat insufficient. Although this assumption may be approximately true in some situations, it is deemed that this assumption in its current form is too strong and simplifies the analysis too much. Consider training a ReLU network; due to non-smoothness, the gradients can change abruptly due to a very small change in the weight value. Hence, I believe that the analysis of anti-correlation needs to be made more rigorous, through an analysis that takes within-epoch updates into account.

Overall, although the paper tackles an important problem and offers good insights, the theoretical analysis is based on strong assumption(s) that is hard to justify. I recommend the authors to revise their theory to remove restrictive assumptions, and I believe doing so will significantly improve the paper. At this time, I recommend rejection.

**Additional Comments On Reviewer Discussion:**

Other than the points mentioned above, noteworthy comments from the reviewers include:
- **Strong Assumption 3 (Hessian and noise covariance commute).** The authors clarified that their analysis extends to the non-commutative case.
- **Writing suggestions.** It was pointed out by oKMV that the quantity correlation time is not sufficiently explained in the paper. I agree with the reviewer and recommend the authors to add an intuitive discussion of the quantity. Also, I concur with Reviewer EdGc’s suggestion to avoid using imprecise statements such as $\approx$ or $\gg$ in formal theorem statements.
- **Relevance to an existing preprint.** It was brought up in the review that there is an existing preprint that delivers similar main messages. Although failure to cite this preprint did not affect my recommendation, the authors should contextualize their contributions relative to it in the paper.

---

### Decision · Program_Chairs · 2025-01-22

Reject